

# Can liquid cloud microphysical processes be used for vertically-pointing cloud radar calibration?

Maximilian Maahn[1,2], Fabian Hoffmann[1,3], Matthew D. Shupe[1,2], Gijs de Boer[1,2], Sergey Y. Matrosov[1,2], and Edward P. Luke[4]

[1]Cooperative Institute for Research in Environmental Sciences (CIRES), University of Colorado Boulder, Boulder, Colorado, USA
[2]NOAA Earth System Research Laboratory (ESRL), Physical Sciences Division, Boulder, Colorado, USA
[3]NOAA Earth System Research Laboratory (ESRL), Chemical Sciences Division, Boulder, Colorado, USA
[4]Brookhaven National Laboratory, Upton, New York, USA

**Correspondence:** Maximilian Maahn (maximilian.maahn@colorado.edu)

**Abstract.** Cloud radars are unique instruments for observing cloud processes, but uncertainties in radar calibration have frequently limited data quality. Thus far, no single, robust method exists for assessing calibration of past cloud radar data sets. Here, we investigate whether observations of microphysical processes of liquid clouds such as the transition of cloud droplets to drizzle drops can be used to calibrate cloud radars. Specifically, we study the relationships between the radar reflectivity

factor and three variables not affected by absolute radar calibration: the skewness of the radar Doppler spectrum ($\gamma$), the radar mean Doppler velocity ($W$), and the liquid water path (LWP). We identify reference points of these relationships and evaluate their potential for radar calibration. For $\gamma$ and $W$, we use box model simulations to determine typical radar reflectivity values for these reference points. We apply the new methods to observations at the Atmospheric Radiation Measurement (ARM) sites North Slope of Alaska (NSA) and Oliktok Point (OLI) in 2016 using two 35 GHz Ka-band ARM Zenith Radars (KAZR).

For periods with a sufficient number of liquid cloud observations, we find that the methods are robust enough for cloud radar calibration, with the LWP-based method performing best. We estimate that in 2016, the radar reflectivity at NSA was about $1\pm1$ dB too low, but stable. For OLI, we identify serious problems with maintaining an accurate calibration including a sudden decrease of 5 to 7 dB in June 2016.

## 1 Introduction

Due to their profiling capabilities, millimeter cloud radars are one of the most important tools for cloud remote sensing. Their measurements are used for process studies as well as for long term monitoring of cloud and precipitation properties. Although maintaining an accurate radar calibration is absolutely crucial to avoid biases and false trends in observational data sets, calibrating cloud radars accurately is a challenging and long-standing problem. In this study, we tackle this problem and investigate the potential for using observations of liquid cloud microphysical processes for radar calibration.

Radar calibration is quantified by the *radar calibration constant*. Despite the name *constant*, the constant can actually change due to the aging of components, temperature fluctuations, or hardware defects. Therefore, we have to determine not only the



initial calibration constant of a system, but also monitor the calibration constant for changes. For example, waveguide-corrosion of the *MilliMeter wavelength Cloud Radar* (MMCR) of the *US Department Of Energy* (DOE) *Atmospheric Radiation Measurement* (ARM) program at the *North Slope of Alaska* (NSA) site in Utqiaġvik (Barrow), Alaska caused a 9.8 dB calibration offset in 2008 (Protat et al., 2011). Also, a liquid film on the radome or radar antenna caused by precipitation can temporarily

lead to up to 4 dB additional two-way attenuation (Frech, 2009). From an engineering perspective, radar calibration is complicated by the fact that radar returns span several orders of magnitude in power and—particularly for pulsed radars—the span between transmitted and received power is even larger.

The community has multiple approaches to calibrate cloud radars, but none are applicable to all situations. Most commonly, a budget calibration is done wherein all components are calibrated separately and the individual calibration constants are

summed (Chandrasekar et al., 2015). A budget calibration can also be combined with a receiver calibration by observing a reference target emitting microwave radiation. For example, Whiton et al. (1977) proposed pointing a scanning radar into the sun and Küchler et al. (2017) used liquid nitrogen—similar to the standard calibration method of microwave radiometers. Yet, the errors of the individual budget calibrations sum up, and there is a risk of overlooking error sources, e.g. due to interaction between radar components. Therefore, it is advantageous to calibrate the full radar system end-to-end. Atlas (2002) provided

an extensive overview of different end-to-end radar calibration techniques, most of them relying on observing objects with known radar cross-sections. These reference targets included corner reflectors and various metallic or metalized spherical objects such as ping pong balls, ball projectiles from air guns, and Christmas ornaments. However, observations of reference targets require dedicated field operations and cannot be used to calibrate past data sets. Also, the observation of a reference target with a radar can be challenging for a number of reasons. First, most reference targets do not move, and hence do not

cause a Doppler shift so that the target's return cannot be distinguished from ground clutter unless the target is positioned far away from the surface. For lifting the target from the surface, past studies proposed using fiberglass poles (Kollias et al., 2016), tethered balloons (Atlas and Mossop, 1960), or unmanned aerial vehicles (Küchler et al., 2017). Second, the exact location of the target with respect to the radar needs to be known for calibration because reference targets are point targets. Instead, atmospheric hydrometeors are volume distributed targets. Third, the antenna properties are not well defined unless the target

is in the antenna's far-field, i.e., at least a couple of hundred meters away from the radar. Lastly, receiver saturation must be avoided, which requires the use of an attenuator or a sufficient distance between the calibration target and the radar. Because of these reasons, calibration by reference targets is only feasible for scanning cloud radars, but not for vertically pointing cloud radars, which are most commonly used. Several studies have suggested calibrating radars by comparing their measurements of rainfall with integrated drop size distributions from ground-based disdrometers (Joss et al., 1968; Ulbrich and Lee, 1999; Frech

et al., 2017). Tridon et al. (2017) proposed to use self-consistency checks of retrievals from simultaneous radar observations at multiple frequencies to identify calibration problems. However, disdrometers and regular liquid precipitation are required for monitoring calibration continuously and the challenge of radome or antenna attenuation during precipitation events needs to be considered, particularly for vertically pointing systems.

If multiple radars are available, it is easier to achieve a relative calibration by cross-calibration. Cross-calibration also works

when the radars have different frequencies, as long as the hydrometeors are small enough to assume Rayleigh scattering and





differential attenuation is accounted for (Hogan et al., 2000; Kneifel et al., 2015; Ewald et al., 2018). If the radars are not collo-cated the cross-calibration can also be done statistically by comparing long-term data sets. But such comparisons can be biased by different radar sensitivities and it is important to degrade both radars to the same sensitivity. Protat et al. (2011) compared observations statistically from the CloudSat satellite W-band radar with ground-based observations for relative calibration. Be-

cause CloudSat's calibration is well established (Tanelli et al., 2008), Protat et al. (2011) and Louf et al. (2019) proposed to use CloudSat as a reference for an absolute calibration of ground-based radars. However, long time series of at least several months are required (Kollias et al., 2019) and the method cannot be used to monitor radar calibration at higher temporal resolutions. Merker et al. (2015) proposed another method for absolute radar calibration of radars by inter-comparisons but their method requires a very specific setup with three small radars.

We can also avoid the problem of absolute radar calibration by using variables not affected by absolute calibration such as the higher moments of the radar Doppler spectrum (Maahn et al., 2015), attenuation (Matrosov, 2005) and some polarimetric variables such as depolarization ratio (Matrosov et al., 2017), differential reflectivity, and differential phase shift (Oue et al., 2018). Yet, excluding variables reduces the information content of the observations significantly (Maahn and Löhnert, 2017), depending on the application.

In summary, no method for obtaining an absolute calibration is available that works in all situations. Either dedicated field campaigns or in-situ observations of drop size distributions are required. Budget calibrations are not end-to-end, and relative calibrations require trusting the calibration of a reference radar. In order to close this gap, we investigate whether liquid cloud microphysical processes can be used for radar calibration. Luke and Kollias (2016) proposed to use the unique relationship between the *equivalent radar reflectivity factor* (here *reflectivity* or $Z_e$, in dBz, Smith, 2010) and the *skewness* of the radar

Doppler spectrum ($\gamma$, unitless) during drizzle-onset—commonly defined as drops exceeding the critical diameter for starting autoconversion (20 to 40 $\mu$m)—for calibration. Several studies have suggested that $\gamma$ is helpful for studying drizzle formation (Kollias et al., 2011a, b; Luke and Kollias, 2013; Acquistapace et al., 2019). Further, Luke and Kollias (2016) suggested that the relationship between the *liquid water path* (LWP, in kg m$^{-2}$) and the maximum reflectivity in the column max($Z_e$) contains information that can be used for radar calibration. LWP and max($Z_e$) are correlated because larger LWP values permit drops

to grow larger by condensation and enhance the probability of drizzle formation leading to higher $Z_e$ values (see Fig. 1 of Acquistapace et al., 2019). In this study, we evaluate whether the $Z_e$ - $\gamma$ and LWP- max($Z_e$) relationships can be used for calibrating vertically pointing cloud radars. In addition to these two relationships proposed by Luke and Kollias (2016), we also investigate the relationship between $Z_e$ and the *mean vertical Doppler-velocity* ($W$, in m s$^{-1}$), because $W$ has been successfully used for drizzle detection (e.g., Shupe, 2007) due to the larger fall-velocity of drizzle drops.

We run box model simulations of drizzle-onset to develop the details of the method, characterize its uncertainties, and apply it to radar observations of the North Slope of Alaska (NSA) and Oliktok Point (OLI) ARM sites from 2016. The instruments, data sets, box model, and radar simulator used in this study are detailed in section 2. The calibration methods used in this study are presented in section 3. Besides the three new methods based on liquid cloud microphysical processes, we use a reference method to calibrate the two cloud radars relative to one another. For this, we modify the relative calibration method

which Protat et al. (2011) proposed for calibrating ground-based cloud radars with CloudSat. In section 4, we apply the various



**Table 1.** Technical specifications of the radars in Oliktok Point and Utqiaġvik (Barrow)

|  | Oliktok Point | Utqiaġvik (Barrow) |
|---|---|---|
| Abbreviation | OLI | NSA |
| Radar | KAZR2 | KAZR |
| Frequency [GHz] | 34.83 | 34.83 |
| Mode | general (ge) | general (ge) |
| FFT points [-] | 512 (256)* | 256 |
| Pulse repetition frequency [Hz] | 2771.31 | 2771.31 |
| Spectral averages [-] | 9 (18) | 20 |
| Dwell time [s] | 1.69 | 1.85 |
| Nyquist velocity [m s$^{-1}$ ] | 5.977 | 5.963 |
| Sensitivity at 1 km [dBz] | $-37.3$ ($-39.0$) | $-32.7$ |

* Specifications in parenthesis correspond to the configuration before 2016-06-16

calibration methods to data from NSA and OLI and assess the temporal evolution of the calibration quality at both sites. We quantify the calibration quality using the calibration offset $O$ defined as

$$Z_e^{\text{truth}} = Z_e^{\text{measured}} + O. \tag{1}$$

Finally, concluding remarks are given in section 5.

## 2   Data sets and models

### 2.1   Sites

In this study, we use ground based remote sensing observations from two observatories operated by the DOE ARM Program located in northern Alaska: Utqiaġvik (ARM's North Slope of Alaska (NSA) site, formerly known as Barrow, 71.323°N, 156.616°W) and Oliktok Point (OLI, 70.495°N, 149.886°W). While the former was established in 1996, the latter did not become fully operational until late 2015. Both sites are located on the coast of the Beaufort Sea and lie only 250 km apart. The synoptic-scale forcing is very similar, resulting in high correlations between both sites for sea level pressure and near-surface air temperature, humidity, and wind (Maahn et al., 2017).

### 2.2   Instruments and observations

Both sites are equipped with a 35 GHz Ka-band ARM Zenith Radar (KAZR). While the radar at NSA is a first generation KAZR, the one at OLI is a second generation KAZR2 with improved sensitivity (Table 1). The spectral resolution of the OLI KAZR2 was increased from 256 to 512 Doppler spectral bins on 2016-06-16. For the radar moments $Z_e$, $\gamma$, and $W$,





we use the radar product presented in Williams et al. (2018), which, unlike the standard ARM general mode (GE) moment products, includes an advanced clutter removal and higher moments such as $\gamma$. Because turbulence can mask microphysical signals in $\gamma$, observations with high temporal resolution are usually required for minimizing broadening effects of the Doppler spectrum (Acquistapace et al., 2017). Instead, Williams et al. (2018) use a shift-then-average method to reduce the impact of

turbulence on the radar moments allowing the use of coarser temporal resolution (15 s). For temperature and humidity profiles, we use the standard ARM interpolated radiosonde product (ARM user facility, 1999, updated daily) based on three (two) daily launches at NSA (OLI). Further, both sites are equipped with ceilometers for cloud base estimation (Vaisala CL31, ARM user facility, 1996, updated daily) and microwave radiometers (MWR) to retrieve LWP and integrated water vapor (IWV) using the MicroWave Radiometer RETrieval (MWRRET) algorithm (Turner et al., 2007). To minimize MWR retrieval biases, we

applied monthly offset corrections to the observed brightness temperatures using clear-sky radiosonde observations. At NSA, we estimate LWP from a combination of the 90 GHz channel of an RPG-150-90 radiometer (ARM user facility, 2006, updated daily, the 150 GHZ channel was not operational in 2016) and the 23.8 GHz and 31.4 GHz channels of a Radiometrics WVR-1100 radiometer (ARM user facility, 1993, updated daily). At OLI we retrieve LWP from a three channel (23.834, 30, and 89 GHz) Radiometrics PR2289 radiometer (ARM user facility, 2011, updated daily). For identifying cloud phase, we use the

phase classification by Shupe (2007), which depends on a combination of KAZR, MWR, radionsondes and micropulse lidar (MPL,  ARM user facility, 1990, updated daily) measurements.

The site at OLI was also equipped with a Ka-band Scanning ARM Cloud Radar (KaSACR) from March 2016 to September 2017. However, the KaSACR was pointing vertically for only 10 minutes per hour. Combined with its reduced sensitivity, this leads to too few observations of liquid clouds and we decided not to include KaSACR observations in this study.

Unless stated otherwise, $Z_e$ is corrected for gaseous attenuation (Rosenkranz, 1998) using the radiosonde profiles scaled by the MWR's IWV. Two-way integrated gaseous attenuation is typically less than 0.4 dB for the whole vertical column at Ka-band. Attenuation by liquid water is neglected. $W$ is adjusted to sea level air density following Zawadzki et al. (2005).

We analyze observations of the full year 2016 obtained at both sites. The time period was selected because the KAZR at OLI became fully operational only in fall 2015 and suffered from a malfunction of a phase lock oscillator resulting in

resonance peaks in the Doppler spectrum for most of 2017. Even though $Z_e$ and $W$ should not be affected by this, the full Doppler spectrum is required by the clutter removal method of Williams et al. (2018). At OLI, clutter removal is essential for investigating low liquid clouds with low signal to noise ratio (SNR).

## 2.3  Box model

To simulate the transition from cloud droplets to drizzle drops in an idealized way, we use a zero-dimensional box model of

the droplet collection process (Hoffmann et al., 2017). The box model's results will allow us to determine the potential of using drizzle onset for radar calibration. The box model is based on the *superdroplet* approach, in which several hundred computational particles are simulated, each superdroplet representing an ensemble of real, identical droplets. We apply the so-called *all-or-nothing* approach to calculate collections among the superdroplets and the *singleSIP* initialization method.



Unterstrasser et al. (2017) showed that both methods are preferable to represent collision-coalescence in the superdroplet modeling framework.

The model was initialized with 500 superdroplets, whose sizes and weighting factors (the number of real droplets represented by a superdroplet) are chosen to represent a lognormal drop size distribution (Feingold and Levin, 1986)

$$N(D) = \frac{N_{tot}}{\sqrt{2\pi}\ln(\sigma_g)D}\exp\left[\frac{-\ln^2(D/d_g)}{2\ln^2(\sigma_g)}\right],\tag{2}$$

with $D$ the droplet diameter, $N_{tot}$, the total number of droplets, $d_g$ the geometric mean diameter, and $\sigma_g$ the geometric standard deviation. We vary these parameters systematically in this work to evaluate sensitivity to particular quantities. Collision-coalescence is steered by the collection kernel, in which the droplet velocity difference is calculated using terminal velocities by Beard (1976), the collision efficiencies are taken from Hall (1980), coalescence efficiency is assumed as unity, and turbulent enhancement is described as in Ayala et al. (2008) and Wang and Grabowski (2009). Turbulence enhancement of the collision process is controled by a prescribed energy-dissipation-rate (see Riechelmann et al., 2012). The simulation time has been restricted to 3h. Note that no other microphysical processes besides collision-coalescence are considered, and droplets are not allowed to sediment from the box, i.e. the liquid water content (LWC) remains constant. See Hoffmann et al. (2017) for more details on this modeling approach.

## 2.4 Radar simulator

To convert the drop size distributions (DSDs) of the box model into radar observables, we use the spectral radar simulator of the second generation Passive and Active Microwave radiative TRAnsfer model (PAMTRA2; https://github.com/maahn/pamtra2). Its physical basics are the same as for the first generation PAMTRA (Maahn et al., 2015; Maahn and Löhnert, 2017), but it is designed in a more modular way. Because the drop size in the box model does not exceed 1/10th of the radar wavelength (8.6 mm) for $Z_e < 10$ dBz, we can use Rayleigh scattering for estimating the radar backscattering cross section of the drops. From the backscattering cross-section, the radar Doppler spectrum is estimated using the same fall-velocity-size relationship as in the box model (Beard, 1976). Unlike for $Z_e$ and $W$, broadening by the Doppler spectrum due to turbulence imposing random motion on the droplets needs to be accounted for when estimating $\gamma$. For this, we convolve a Gaussian velocity distribution with the idealized radar spectrum. The standard deviation of the Guassian distribution depends mostly on the degree of turbulence and the contribution of the horizontal wind field to the radial velocity due to the finite radar-beamwidth following (Shupe et al., 2008). The former is estimated from the energy-dissipation-rate $\epsilon$, which is varied as discussed below, and a constant horizontal wind of $10 \text{ m s}^{-1}$ is assumed for the latter. Noise is added to the spectrum in correspondence with KAZR2 specifications after June 2016 (Table 1). From the simulated radar Doppler spectrum we estimate its moments including radar reflectivity $Z_e$, mean Doppler-velocity $W$, and skewness $\gamma$ following Maahn and Löhnert (2017).



## 3 Calibration methods

### 3.1 Skewness and mean Doppler velocity-based methods

We hypothesize that there is a reference point during drizzle-onset that has a typical $Z_e$ value, which can be constrained by $\gamma$ or $W$. To determine these reference points we use and analyze the results of the box model-radar simulator combination introduced above for simulating drizzle-onset. Focusing on the formation of drizzle drops from cloud droplets—referred to as autoconversion—we assume that collision-coalescence is the dominating cloud process during drizzle-onset and that other cloud processes can be neglected for this purpose. To assess the model's sensitivity to the microphysical properties of a given cloud, we first vary the initial DSDs (Sect. 3.1.1). Based on these results, we determine the best reference points for radar calibration (Sect. 3.1.2) and discuss how to apply these reference points to observations (Sect. 3.1.3).

### 3.1.1 Sensitivity study

We chose a set of parameters featuring a slow cloud-to-drizzle transition in agreement with observations of DSDs (Geoffroy et al., 2010) and turbulence (Shupe et al., 2012; Maahn et al., 2015): $N_{tot} = 10^8$ m$^{-3}$ as the initial drop number, $\sigma_g = 1.34$ as the standard geometric deviation, $d_g = 1.6 \times 10^{-5}$ m as the geometric mean diameter to describe the initial lognormal distribution (eq. 2), and $\epsilon = 10^{-4}$ m$^2$ s$^{-3}$ as the turbulent energy-dissipation-rate. This DSD corresponds to 0.26 g m$^{-2}$ LWC. We refer to the simulation using these values as the reference run in the following. The results of the reference run show (orange lines Fig. 1) that $Z_e$ increases monotonically and that $\gamma$ reflects the competition of cloud droplets and drizzle drops in the radar Doppler spectrum. The shape of the $Z_e$-$\gamma$ relationship is typical for the cloud droplets to drizzle drops transition (Kollias et al., 2011b). In the absence of drizzle, only backscattering by cloud droplets contributes to the radar Doppler spectrum. For this stage, the Doppler spectrum has a Gaussian shape (i.e., $\gamma \approx 0$), the variability of droplet fall velocities is small, and turbulence regulates the width of the Doppler spectrum. As soon as the first drizzle drops are created by autoconversion after 45 min, the $\gamma$ values become positive (motion towards the radar is defined as positive in this study), because the drizzle drops extend the tail of the Doppler spectrum towards faster, more positive velocities. The maximum $\gamma$ value of approximately 0.7 is reached at -20.2 dBz ($Z_e^{\max(\gamma)}$). The critical droplet diameter required to start autoconversion varies between 14 and 80 $\times 10^{-6}$ m depending on the DSD (Liu et al., 2004). When drizzle and cloud droplets contribute approximately equally to $Z_e$, the shape of the spectrum is again more symmetric resulting in $\gamma \approx 0$. This stage is referred to as the cloud-drizzle balance point in the following and is reached after another 45 min at $-16.5$ dBz ($Z_e^{\gamma=0}$). Finally, $\gamma$ becomes negative when the spectrum is dominated by drizzle drops and the remaining cloud droplets extend the tail of the spectrum to the opposite, smaller-droplet side. However, simulated values significantly larger than $Z_e^{\gamma=0}$ have to be treated with care because drizzle removal from the cloud by sedimentation is not accounted for by the box model.

To assess the sensitivity of the $Z_e$-$\gamma$ relationship to microphysics, the initial parameters of the box model were perturbed. We chose the perturbations such that a realistic range is covered, but made sure that drizzle is created neither instantly nor too slowly (i.e., no drizzle after 3 h runtime). To evaluate the sensitivity with respect to $N_{tot}$, we divided and multiplied $N_{tot}$ by a factor of two (Fig. 1.a). When cloud droplets dominate the radar signal, $N_{tot}$ scales linearly with $Z_e$ in linear units and





the offset between the model runs is close to 3 dB (corresponding to a factor of 2 as expected from the modification of $N_{tot}$). Consequently, the $Z_e$ values for maximum $\gamma$ (referred to as $Z_e^{\max(\gamma)}$ in the following) are approximately 3 dB apart (-23.5, -20.2, and -17.6 dBz). However, autoconversion is more efficient for greater number concentrations (with constant droplet size) so that $\gamma$ decreases faster as a function of both $Z_e$ and time than for the other runs. Due to these compensating effects, $Z_e$ values

for the cloud-drizzle balance point with $\gamma = 0$ ($Z_e^{\gamma=0}$) are closer together (-16.5 and -15.4 dBz) than for the maximum of $\gamma$. Interestingly, this is not the case if we reduce $N_{tot}$ by 50%. Then, the $Z_e$-$\gamma$ line is shifted to the lower left and $Z_e^{\gamma=0}$ is $-21.1$ dBz and $4.7$ dB smaller than for the reference run. For this run, autoconversion is so slow that after 2 h cloud droplets still dominate the spectrum and a reflectivity of only -15 dBz is reached at the end of the 3 h simulation. For the run with doubled $N_{tot}$, the time required until the drizzle dominates the radar Doppler spectrum (i.e., $\gamma < 0$) is less than 1 hour.

For estimating the sensitivity to the width of the size distribution, we perturb $\sigma_g$ by $\pm 0.05$ (Fig. 1.b). If we perturbed the initial DSD width by larger values, the box model would create drizzle too slowly or too quickly for our purposes. While the $Z_e^{\max(\gamma)}$ values for both perturbations are about 2 dB apart, the difference between the $Z_e^{\gamma=0}$ values are 2.9 and 0.2 dB for the reduction and increase of $\sigma_g$, respectively. Similar to the doubled $N_{tot}$ run, autoconversion is more efficient and faster when we increase $\sigma_g$. At the same time, a narrower distribution leads to a larger absolute $\gamma$ value due to the reduced Doppler spectrum

width of the cloud peak. Note that the reference run and the run with increased $\sigma_g$ are almost identical for $Z_e > -18$ dBz, but the run with reduced $\sigma_g$ remains different. This highlights that the presence of larger droplets in the initial spectrum (due to a larger standard deviation) is important for drizzle-onset, but the effect saturates when drizzle drops become more numerous. This is similar for the runs where $d_g$ has been increased and reduced by $\pm 1 \mu$m (Fig. 1.c): $Z_e^{\gamma=0}$ changes little when increasing $d_g$ (-16.3 dBz), but is reduced for a smaller $d_g$ (-19.4 dBz).

Turbulence also impacts drizzle-onset. $\epsilon$ is perturbed by one order of magnitude in agreement with the range of observations of Arctic clouds by Shupe et al. (2012) (Fig. 1.d). Enhanced turbulence leads to turbulent broadening, which reduces the $\gamma$ magnitude by making the spectrum more symmetrical (Acquistapace et al., 2017). This is particularly visible for low reflectivities, which are dominated by cloud droplets. Turbulence has only a small impact on autoconversion, which can be seen by the slightly faster drizzle formation and the small change in $Z_e^{\gamma=0}$ of 0.2 dB. Similar results have been found in other simulations

by Hoffmann et al. (2017), in which turbulence did not significantly change the timing of drizzle, but rather the amount of cloud water transformed to drizzle.

In reality, a change in $N_{tot}$ alone is not very realistic, because when $N_{tot}$ is e.g., increased, the available liquid is typically distributed on a larger number of smaller sized droplets. In other words, an increase in $N_{tot}$ for fixed LWC, which would shift the $Z_e$ - $\gamma$ relationship towards larger $Z_e$ values, is compensated by a reduction of $d_g$, which would shift the relationship to

the opposite direction. To investigate this, we repeated the $N_{tot}$ variation for fixed LWC by changing $d_g$ accordingly (Fig. 1.e). Note that the required change in $d_g$ is larger (18.9 and 11.9 $\mu$m) than investigated above. As expected, autoconversion is more efficient in the low $N_{tot}|_{\text{LWC}}$ case, but there is apparently an upper threshold for $Z_e^{\gamma=0}$, which increases only by 1 dB. For the high $N_{tot}|_{\text{LWC}}$ case, $Z_e^{\gamma=0}$ is reduced strongly from -16.5 to -22.9 dBz. Unlike for the other runs, drizzle formation is very slow and droplets still dominate after 2 hours of model run time. Interestingly, the steeper slope of the $Z_e - \gamma$ relationship for the





high $N_{tot}|_{\text{LWC}}$ case is in agreement with the results of Kollias et al. (2011a) who compared maritime and continental (implying higher $N_{tot}$ values) datasets.

Collision-coalescence including autoconversion is a stochastic process so a random number generator is used in the box model for emulation. To make sure the runs are comparable, we previously seeded the random number generator with the same
number for the sensitivity study. Here, we use five different seeds for the reference initial DSD in order to quantify the role of chance. Figure 1.f shows that the impact is surprisingly high and $Z_e^{\gamma=0}$ ($Z_e^{\max(\gamma)}$) varies between $-16.5$ and $-18.9$ dBz ($-20.0$ and $-21.6$ dBz). We conclude from this that the stochastic nature of collision-coalescence reduces the impact of the clouds' initial DSD on the $Z_e - \gamma$ relationship. However, the impact of stochasticity is likely overestimated in the box model because of the limited number of simulated superdroplets (Dziekan and Pawlowska, 2017).
Because the drop fall-velocity depends strongly on size, $W$ increases with increasing drizzle concentration. Consequently, we also evaluate the results of the sensitivity study with respect to the $Z_e$ - $W$ relationship (Fig. 2) for comparison. On the one hand, $W$ is more prone to biases than $\gamma$, e.g., due to radar miss-pointing or vertical air motions. While we assume that the latter cancels for longer time series, consistent lifting related to orography could bias $W$ even for long-term data sets. On the other hand, $W$ doesn't require high temporal resolution of the radar (Acquistapace et al., 2017) and is less noisy as can also be seen
from the spread of the results (purple dots). The dependence on the initial DSD is similar to the $Z_e$ - $\gamma$ relationship. The fact that drizzle develops more efficiently for DSDs with larger $N_{tot}$, $\sigma_g$, or $d_g$ can be seen from the slower $W$ for the same $Z_e$. This is because $W$ (proportional to the first DSD moment for drizzle) increases more slowly with size than $Z_e$ (proportional to the sixth DSD moment). Note that $W$ does not depend on $\epsilon$, therefore the runs with different $\epsilon$ are practically identical. Note that, unlike for $Z_e^{\gamma=0}$, there are apparently no saturation effects limiting the variability of the $Z_e - W$ relationship.

**3.1.2   Determining reference values**

Using the box model, we investigate whether the $Z_e$ - $\gamma$ or $Z_e$ - $W$ relationships have the potential to be used for radar calibration. When using observations over a longer time period, we cannot assume that the same initial conditions prevail for the whole time. Therefore, we combined all perturbations of $N_{tot}, \sigma_g, d_g$, and $\epsilon$ with each other in order to cover the parameter space of initial conditions better than for the sensitivity study. Every run was repeated five times with different assigned seeds
(i.e. $5 \times 3^4 = 405$ runs). To make sure the full cloud droplet to drizzle transition is included in the data set, only runs without drizzle at model initialization are considered. Also, runs without any drizzle production within 3 h are omitted, which leaves 340 runs.

The results show considerable spread (Fig. 3). To obtain the median relationship for $\sigma$ and $W$, we bin the data by $Z_e$ (bin width 1 dB) and estimate the median values of $\gamma$ and $W$ for every bin. We smooth the resulting curve using the Savitzky-
Golay filter (window length 7, polynomial order 2, Savitzky and Golay, 1964). That is particularly important when applying the method to observations (see below), because it makes the method more robust by increasing the number of observations contributing to a particular point on the curve. Typically, the smoothing changes $Z_e^{\gamma=0}$ by less than 1 dB. The resulting median relationships show the typical partly sinusoidal-shaped $Z_e$ - $\gamma$ relationship and an increase in $W$ for $Z_e > -20$ dBz. We maintain that this median curve is much better suited for calibration, because the mean reflectivity would be more sensitive to



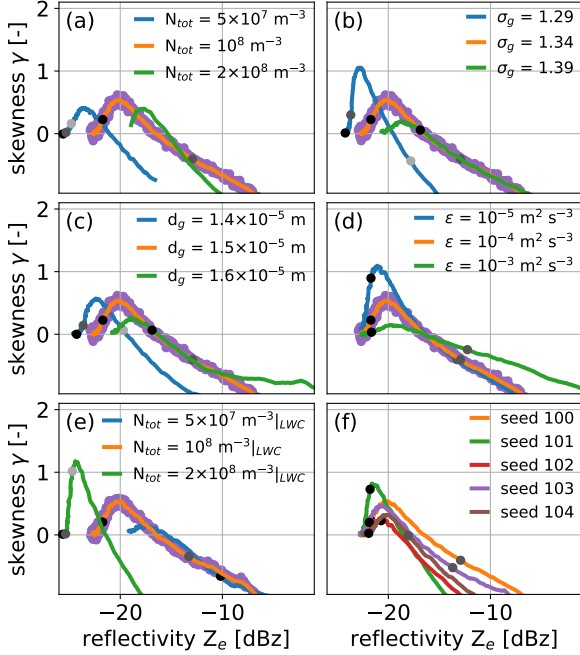

**Figure 1.** Sensitivity of the reflectivity $Z_e$ - skewness $\gamma$ transition for drizzle-onset to (a) total number concentration $N_{tot}$, (b) the unitless standard deviation of the log-normal distribution $\sigma_g$, (c) the geometric mean diameter $d_g$, and (d) the turbulent energy-dissipation-rate $\epsilon$. We also (e) modified $N_{tot}$ while keeping liquid water content (LWC) constant (i.e. increasing $d_g$) and (f) used different seeds for the box model. The purple points show all data points of the reference run, the lines denote smoothed model results. The black, dark gray, and light gray indicate model simulation times of 1 h, 1.5 h, and 2 h, respectively. Note that the orange lines are identical for all panels.

outliers. A certain $\sigma$ value is not unambiguous and, e.g., a value of $\gamma = 0$ can also refer to a spectrum consisting only of cloud droplets. Therefore, it is important to consider the whole $Z_e$-$\gamma$ relationship instead of determining a mean value for all $Z_e$ with $\gamma = 0$.

    To determine which reference point is best suited for calibration, we estimate the uncertainties of several reference values

5  for $\gamma$ (maximum, $0, -0.1$) and $W$ $(0.25, 0.5, 0.75$ m s$^{-1})$ for comparison. The choice of the reference values is somewhat arbitrary but the variability increases strongly outside the investigated range of reference values, which enclose the onset of drizzle. While the determination of max($\gamma$) is straightforward, we estimate the other values by linear interpolation from the neighbouring $Z_e$ bins. In case a reference point is crossed more than once by the median relationship (e.g., $\gamma$ of cloud droplets is also close to zero), we choose the crossing associated with larger $Z_e$ value. The use of the Savitzky-Golay filter ensures

10  that adjacent $Z_e$ bins impact the reference values, which makes the method more stable. Unlike other $Z_e$ calibration studies (e.g., Protat et al., 2011), we do not need to account for radar sensitivity differences, because the range of relevant $Z_e$ values is strictly limited and well above the sensitivity limit. To assess the uncertainty of the reference values, we use a bootstrapping





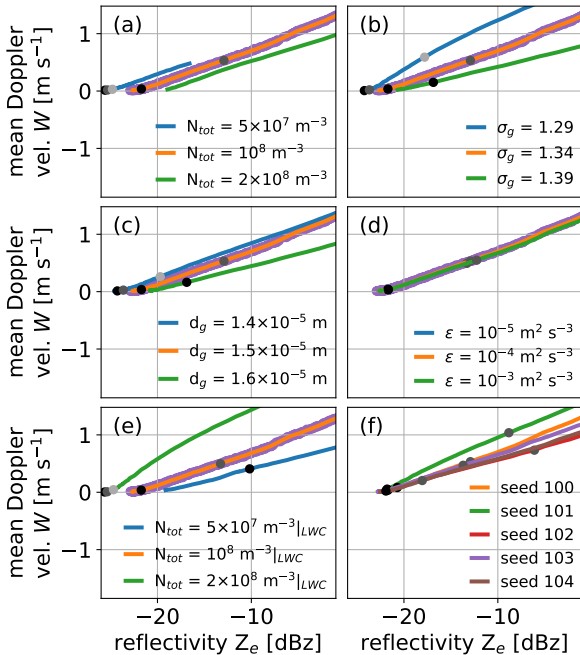

**Figure 2.** As Fig. 1, but for mean Doppler-velocity $W$.

approach: We select 5% of the 340 runs randomly 100 times and determine the resulting reference values for each subset. We estimate the final reference values and their uncertainties from the means and standard deviations, respectively (see uncertainty bars in Fig. 3). For $\gamma$, the comparison reveals that the variability of $Z_e$ is less for reference $\gamma$ values 0 and $-0.1$ ($\pm$ 0.7 and 0.8 dB, Table 2) than for the maximum of $\gamma$ ($\pm$ 1.6). For $W$, the variability is generally larger ($\pm$ 0.8 to 1.9 dB).

Even though we chose the initial conditions to be representative of liquid stratiform clouds at high latitudes, it is possible that our choice of initial conditions is biased. Therefore, we repeated the box model experiment with initial conditions based on aircraft in situ observations from the same region as the cloud radars expecting that measured DSDs include all microphysical processes including advection and sedimentation (Fig. 4). For this we use data of the 5th ARM Airborne Carbon Measurements (ACME-V) aircraft campaign. This campaign took place from June to September 2015 and included cloud probe observations

near the North Slope of Alaska (ARM user facility, 2016). Here, we use liquid-only cloud observations in the vicinity of OLI and NSA. We use every 10th profile of the data shown in the Figs. 4.a and 4.b of Maahn et al. (2017). Except for the initial DSDs, the setup is identical to the idealized runs introduced above. $\epsilon$ was not measured during ACME-V and we apply the same $\epsilon$ values as for the sensitivity study to each measured profile ($\epsilon = 10^{-3}, 10^{-4}$, and $10^{-5}$ m$^2$ s$^{-3}$). Every run was repeated 5 times with different seeds; runs that do not produce drizzle or that include drizzle in the initial DSD are not considered. By

doing so, we avoid the impact of potential sampling problems of large, rare drizzle drops by the in situ probes. This leaves 237 runs and the bootstrapping method is used to determine the uncertainties of the reference points. Even though the estimated





$Z_e$ - $\gamma$ and $Z_e$ - $W$ relationships are more uneven, the general shape between -20 and -10 dBz is very similar to the runs using lognormal DSDs (Table 2).

Note that the minimum required $Z_e$ for drizzle formation is different for ACME-V than for the idealized DSDs, because Figs. 3 and 4 contain only box model runs where drizzle eventually formed. While for the idealized DSDs, drizzle is formed
only when $Z_e$ of the initial DSD is at least -27 dBz, drizzle can form at less than -30 dBz for the ACME-V DSDs. We relate this to the non-idealized nature of the initial ACME-V DSDs and the fact that a single, larger cloud droplet can trigger drizzle formation if included in the observed DSD. As a side effect, the drizzle formation at lower $Z_e$ values leads to enhanced $\gamma$ values below $-20$ dBz. This is most likely only a spin-up effect of the box model, which can be seen from the excellent agreement of the median curves for larger $Z_e$. Note that also at $-21$ dBz, $\gamma$ is around zero due to competition between runs with higher and
lower $\gamma$ values. But this does not bias the method because we only use the crossing with the largest $Z_e$ value.

For both initial DSDs, the uncertainties determined from bootstrapping are minimal for $\gamma = 0.0$ and $W = 0.25$ m s$^{-1}$ and we conclude that $Z_e^{\gamma=0}$ and $Z_e^{W=.25}$ are the best reference values for assessing radar calibration. Initializing the simulations with the lognormal and ACME-V DSDs, $Z_e^{\gamma=0}$ is $-17.3 \pm 0.7$ and $-17.8 \pm 1.2$, respectively (Table 2) . $Z_e^{W=.25}$ is estimated as $-16.3 \pm 0.8$ and $-16.9 \pm 1.5$ dBz, respectively. Combining both set ups, we obtain $Z_e^{\gamma=0} = -17.6$ dBz and $Z_e^{W=.25} = -16.6$
dBz. These values are very close to the value of $-17$ dBz proposed by Frisch et al. (1995) for distinguishing between drizzle-free and drizzle containing clouds. Given the idealized set up, we likely underestimated the uncertainties of $Z_e^{\gamma=0}$ and $Z_e^{W=.25}$ and estimate the uncertainty to be at least 3 dB (i.e., a factor of two larger ).

While it is true that we found a much larger variability of $Z_e^{\gamma=0}$ and $Z_e^{W=.25}$ for the sensitivity study (Sect. 3.1.1), we are confident that the reference values can still be determined with sufficient accuracy. We base this claim on the assumption that
observations with reflectivities corresponding to drizzle-onset ($-20$ to $-15$ dBz) are likely dominated by clouds that produce drizzle slowly. Clouds with faster drizzle production reach larger reflectivities quickly, likely have a shorter lifetime, and do not contribute to the data set quantitatively. Clouds without or with extremely slow autoconversion rates will likely not reach $Z_e$ values larger than $-20$ dBz before the end of their lifetime. Together with the significant role of random effects, this indicates that the variability of the $Z_e$ - $\gamma$ and $Z_e$ - $W$ relationships for larger datasets is lower than estimated in the sensitivity study.
By binning the box model results by $Z_e$ and determining the median $\gamma$ and $W$ values, we ensure that slow drizzle generating clouds also dominate our box model estimates, because, similar to observations, clouds forming drizzle quickly in the box model also have quickly increasing $Z_e$ values. Therefore, these clouds contribute little to the reflectivity bins between $-20$ to $-15$ dBz.

This does not mean that $Z_e^{\gamma=0}$ and $Z_e^{W=.25}$ can be used to identify individual profiles with or without drizzle. As shown in
the sensitivity study above and in Acquistapace et al. (2019), the variability from profile to profile can be substantial and $Z_e$ is not suited to identify the presence of drizzle for individual profiles.

### 3.1.3    Application to observations

To determine $Z_e^{\gamma=0}$ and $Z_e^{W=.25}$ from observations, we only use clouds identified by the Shupe (2007) method as purely liquid (corresponding to the phase classes *liquid*, *drizzle*, *liquid+drizzle*, and *rain*) throughout the column. We limit our analysis to





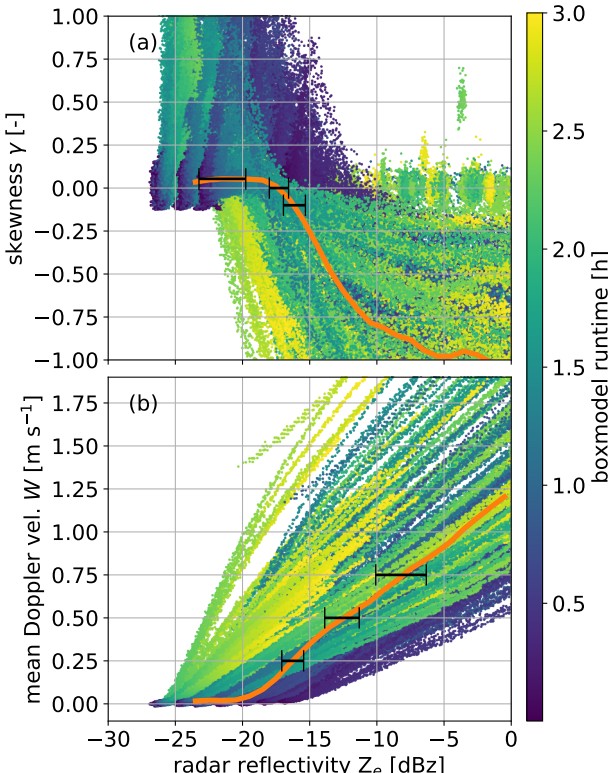

**Figure 3.** (a) reflectivity $Z_e$ skewness $\gamma$ and (b) reflectivity $Z_e$ mean Doppler-velocity $W$ relationships for the individual model runs using synthetic initial model conditions. The orange lines denote the medians as a function of $Z_e$, the black error bars are estimated using bootstrapping for selected $\gamma$ and $W$ values (see Table 2). Color is for model run time.

**Table 2.** Mean $Z_e$ values at various reference points for $\gamma$ and $W$

| Reference Value | idealized $Z_e$ [dBz] | ACME-V $Z_e$ [dBz] |
|---|---|---|
| $\max(\gamma)$ | $-21.3 \pm 1.6$ | $-26.6 \pm 2.0$ |
| $\gamma = 0.0$ | $-17.3 \pm 0.7$ | $-17.8 \pm 1.2$ |
| $\gamma = -0.1$ | $-16.1 \pm 0.8$ | $-16.4 \pm 1.4$ |
| $W = 0.25 \text{ m s}^{-1}$ | $-16.3 \pm 0.8$ | $-16.9 \pm 1.5$ |
| $W = 0.50 \text{ m s}^{-1}$ | $-12.6 \pm 1.3$ | $-14.3 \pm 2.1$ |
| $W = 0.75 \text{ m s}^{-1}$ | $-8.2 \pm 1.9$ | $-11.4 \pm 3.3$ |

observations with cloud base lower than 1000 m and cloud thickness less than 1000 m because we expect that drizzle-onset can be observed best in stratiform clouds due to their lower turbulence. Even though the Doppler spectrum peak identification



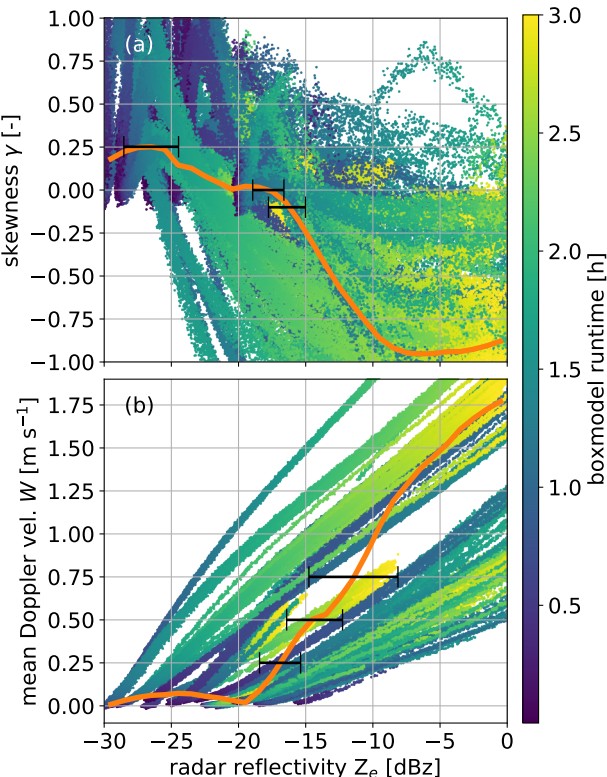

**Figure 4.** As Fig. 3, but using ACME-V observations as initial conditions.

algorithm provided by Williams et al. (2018) can identify atmospheric signals with a SNR as small as -15 dB, we only use data with SNR > -5 dB because $\gamma$ is a particularly noisy variable. $Z_e$ and $W$ are corrected for gaseous attenuation and air density, respectively. We use the same method to estimate $Z_e^{\gamma=0}$ and $Z_e^{W=.25}$ from the observations as from the box model: The liquid cloud observations are binned by $Z_e$ (1 dB bin width), the median $\gamma$ and $W$ values are estimated for each bin, and the resulting curve is smoothed using the Savitzky-Golay filter. Bins with less than 100 observations are omitted from the analysis. To obtain $Z_e^{\gamma=0}$ and $Z_e^{W=.25}$, the median relationships are interpolated linearly.

## 3.2 Liquid water path-based method

While the relationships between $\gamma$ or $W$ and $Z_e$ are shaped by the drizzle-onset process, the correlation between LWP and $Z_e$ is based on the fact that the likelihood of drizzle formation (i.e., increased $Z_e$ values) increases with increasing LWP. But there is also a correlation between LWP and $Z_e$ for non-drizzling clouds: cloud droplets can grow larger in deeper clouds with greater LWP values. Frisch et al. (1998) showed that for non-drizzling, adiabatic clouds with constant $N_{tot}$, LWP is proportional to the cloud-integrated linear square root of radar reflectivity, LWP $\propto \sum_i z_i^{1/2}$ with $z_i = 10^{Z_{e,i}/10}$ for range gate $i$. While this





relationship could be exploited for radar calibration assuming a fixed $N_{tot}$ value, we were not able to apply the method to our data set successfully. This is likely due to challenges identifying a sufficient number of clouds fulfilling the conditions of the method (i.e. non-drizzling, adiabatic clouds with constant $N_{tot}$). Instead, we decided to use the maximum $Z_e$ value in the column ($\max(Z_e)$) to combine the one dimensional LWP with the two dimensional, range-resolved $Z_e$ measurements. Not

relying on the relationship found by Frisch et al. (1998) allows us to not distinguish between non-drizzling and drizzling clouds and use the very same data set as for the $\gamma$ and $W$ methods. Even though $\max(Z_e)$ is likely more noisy than, e.g., the mean of $Z_e$ in the column, $\max(Z_e)$ has the major advantage that the maximum is less likely impacted by radar sensitivity than the mean because a truncation of a distribution's lower end does not impact its maximum. Similar to the $Z_e$-$\gamma$ and $W$ relationships and as shown for cloud-integrated reflectivity by Frisch et al. (1998), the LWP-$\max(Z_e)$ relationship likely also depends

on microphysical (e.g., initial $N_{tot}$) and dynamical (e.g., turbulence, entrainment and mixing) conditions. With respect to the LWP-$\max(Z_e)$ relationship, we expect that higher $N_{tot}$ values lead to reduced $Z_e$ values for the same LWP due to suppression of drizzle formation. But unlike the $\gamma$ and $W$-based methods, which focus on a very specific moment during drizzle-onset, the LWP method is impacted by the full set of processes of droplet growth and drizzle formation, and is potentially impacted by multiple feedback processes between clouds and their environment. For example, the impact of $N_{tot}$ on the LWP-$\max(Z_e)$

relationship would be even larger assuming drizzle suppression due to enhanced $N_{tot}$ leads to larger LWP values (Albrecht, 1989). However, the question of whether and how feedback processes compensate for a LWP increase is still debated (Stevens and Feingold, 2009). Focusing only on drizzle-onset has allowed us to use a simple box model to determine the reference points for the $Z_e$-$\gamma$ and $W$ relationships, but addressing the question of how $N_{tot}$ (and the related cloud condensation nuclei concentration) changes LWP is beyond the scope of this study. Therefore, we decided not to use a model for determining

a reference LWP-$\max(Z_e)$ relationship. Instead, we will use the LWP-$\max(Z_e)$ relationship of one site as a reference and determine the calibration offset of a second site from this. In other words, the LWP-$\max(Z_e)$ relationship is used in a relative way unless we can trust the calibration of one of the radars, which would make it an absolute calibration similar to Protat et al. (2011). Similar to the $\gamma$ and $W$-based methods, this implies the assumption that the LWP-$\max(Z_e)$ relationship is sufficiently stable with respect to changes in microphysical and dynamical conditions. Because we have no box model to identify the LWP

value with lowest variability of $\max(Z_e)$, we do not use a reference point but a reference relationship and minimize the mean weighted difference between reference and observed relationship.

We use the same liquid-only data set as for the $\gamma$ and $W$-based methods, but restrict the observations to cases when the wind direction at cloud level is from the sea. By this, we reduce the potential impact of local pollution at OLI (Maahn et al., 2017; Creamean et al., 2018) which could alter the LWP-$\max(Z_e)$ relationship at OLI due to varying $N_{tot}$. To determine a

LWP-$\max(Z_e)$ for a certain period, we determine mean $\max(Z_e)$ values for LWP intervals of 0.01 kg m$^{-2}$ from 0.01 kg m$^{-2}$ to 0.120 kg m$^{-2}$. For larger LWP values, the number of liquid-only cloud observations drops quickly for the Arctic data set used in this study. When using LWP for radar calibration, it is crucial that the MWR LWP retrievals are offset corrected as discussed in Sect. 2.2.



### 3.3 High altitude clouds method

To evaluate the new methods independently, we apply the relative calibration method based on high altitude clouds proposed by Protat et al. (2011). They estimated a relative calibration offset between CloudSat and ground-based cloud radars by comparing mean reflectivity values of high altitude ice clouds. Here, we adapt this technique to the KAZR dataset of NSA and OLI

assuming that high altitude ice cloud statistics are similar for both sites and have the same mean($Z_e$). This will provide only a relative calibration instead of an absolute one. Comparing mean($Z_e$) of two radars requires that both are limited to the same sensitivity level, therefore we limit the OLI sensitivity to that for NSA. However, changing the relative calibration also changes the difference in sensitivity. To account for this, we implemented the iterative procedure proposed by Protat et al. (2011): after the calibration offset is estimated, the sensitivity limit of the radar at NSA is applied to the OLI radar and the relative calibration

offset is estimated again. This procedure is repeated until the relative calibration offset converges.

    For the comparison, we use all data with—according to radiosondes—an ambient temperature below 0°C above a certain cut-off altitude. To avoid an impact of attenuation from precipitation, profiles containing $Z_e$ values exceeding 10 dBz are discarded. Gaseous attenuation is not accounted for, because both sites are expected to be on average equally affected. The cut-off altitude has to be high enough to avoid local impacts (e.g., due to pollution Maahn et al., 2017) and biases due to individual

frontal systems, but low enough to get a sufficient number of observations. For the latter, we have to consider the low height of the Arctic tropopause in winter. To identify the best cut-off height for every three month period, we apply different cut-off altitudes from 3000 m to 7000 m to the data set and compare two quality control measures. First, we compare the vertical profiles of mean($Z_e$) for OLI and NSA before and after relative calibration. Second, we estimate cloud top altitude statistics, which depend strongly on radar sensitivity (Protat et al., 2011), using 500 m bins before and after calibration. We choose the

cut-off height whose root mean square differences after calibration are best based on both methods.

### 4 Results and discussion

In the following, we apply the four calibration methods introduced above to the data sets of NSA (Fig. 5) and OLI (Fig. 6) in 2016. To investigate temporal trends, we group the data monthly and estimate the calibration offset for every month separately. The only exception is June 2016 because the radar configuration was changed at OLI on June 16th 2016 (see Table

1) potentially affecting radar calibration. Therefore the June data set contains only observations from the first half of June and the remaining observations are combined with the July observations. Due to instrument issues in the second half of June, this affects only a few observations.

### 4.1 Calibration of North Slope of Alaska (NSA) data

For NSA, the monthly $Z_e$ - $\gamma$ relationships follow a sinusoidal-type curve similar to the box model (Fig. 5.a) indicating that

the phase classification is correctly identifying liquid clouds. This is also supported by the fact that all but one month (April 2016) feature $W < 0.2$ m s$^{-1}$ for $Z_e < -25$ dBz as expected for liquid clouds without ice (Fig. 5.e). Also, most monthly





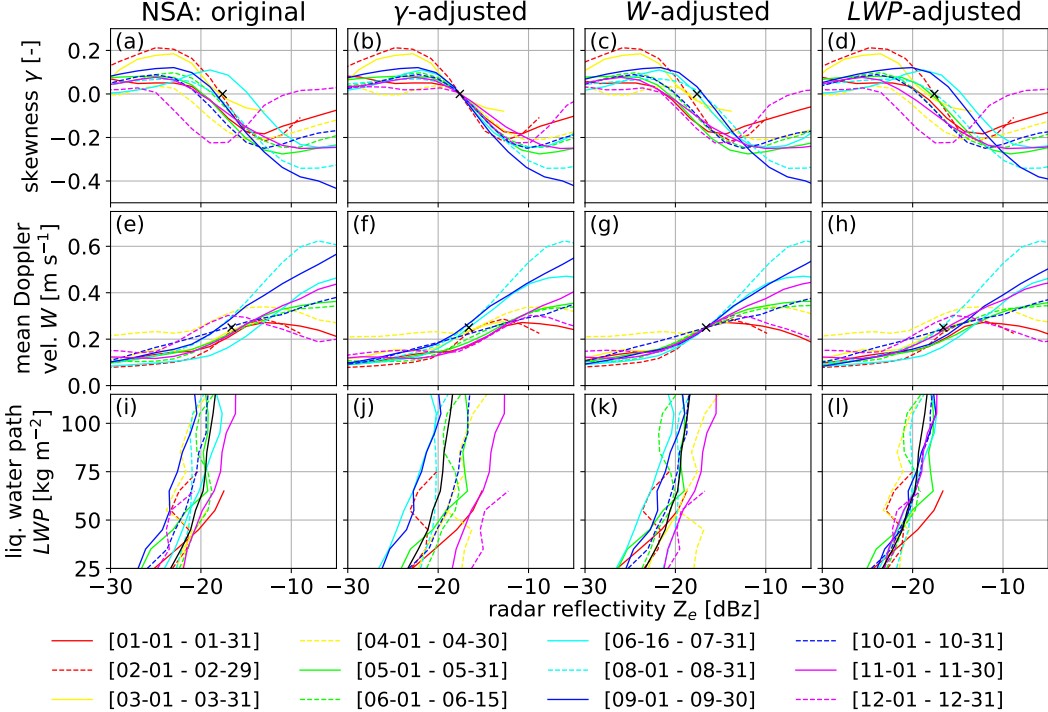

**Figure 5.** Observed reflectivity $Z_e$ - skewness $\gamma$ (a), $Z_e$ - mean Doppler-velocity $W$ (e), and liquid water path LWP-max($Z_e$) (i) relationships for the North Slope of Alaska (NSA). The data has been calibration corrected using the $\gamma$ method (second column), the $W$ method (third column), and the $LWP$ method (fourth column). The colored lines indicate the various calibration periods of 2016. The black cross (rows 1 to 2) shows the reference point used for calibration; the solid black line (row 3) is the reference LWP-max($Z_e$) relationship obtained from the mean of the monthly relationships weighted by the number of observations.

LWP-max($Z_e$) relationships have a similar shape and align within a couple of dB. For the $Z_e$ - $\gamma$ relationship, most monthly relationships have $Z_e^{\gamma=0}$ values between -20 to -17 dBz but there are a couple of outliers. The monthly $Z_e$ - $\gamma$ relationships that are shifted towards smaller (e.g., December 2016) or larger values (e.g., July 2016) show a similar shift for the $Z_e$ - $W$ relationship indicating that both methods are consistent. The shift in the corresponding LWP-max($Z_e$) relationships is smaller,

5  but the December and July relationships are still below and above, respectively, the mean relationship (Fig. 5.i). As discussed above, this, could be related not only to a change in radar calibration, but also to a change in the dominating microphysical conditions. We estimate the calibration offsets $O^{\gamma=0}$ and $O^{W=.25}$ from $Z_e^{\gamma=0}$ and $Z_e^{W=.25}$, respectively, following the definition in Eq. 1 (Fig. 7.a, Table 3). Over the course of the year, $O^{\gamma=0}$ varies between $-2.8$ dB and $7.2$ dB with a mean of $1.8 \pm 2.5$ dB. $O^{W=.25}$ is between $-2.8$ and $3.7$ dB and has a mean of $0.1 \pm 2.0$ dB. Even though there are a couple of outliers enhancing

10  the variability, the standard deviations of 2.5 and 2.0 dB of $O^{\gamma=0}$ and $O^{W=.25}$, respectively, are consistent with the assumed





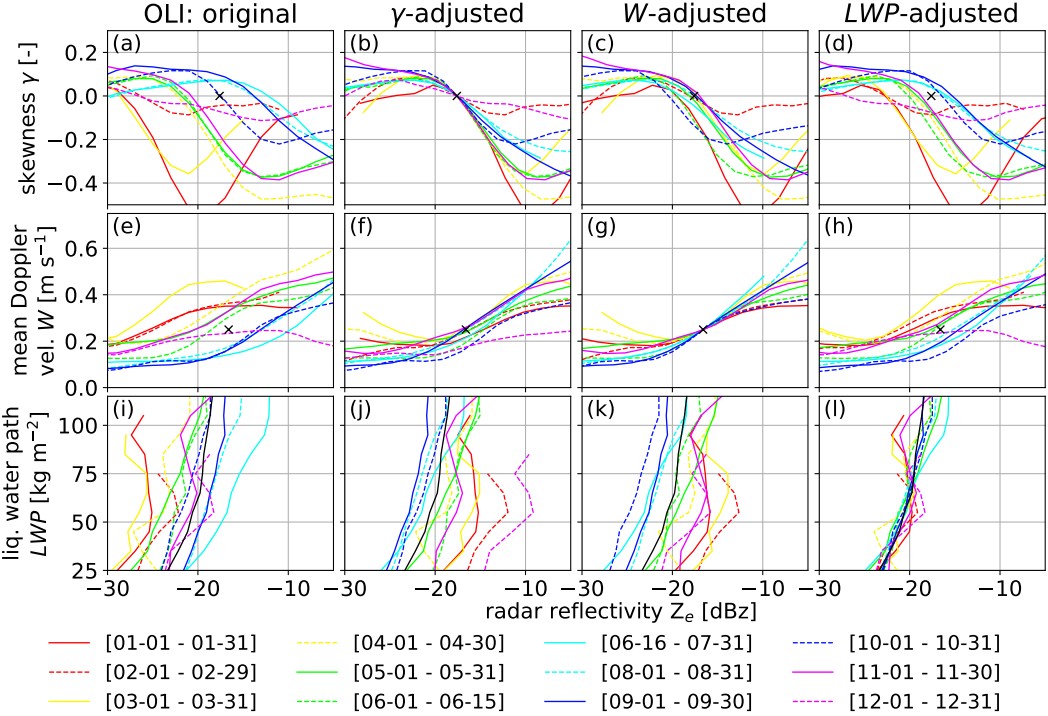

**Figure 6.** As Figure 5, but for Oliktok Point (OLI).

uncertainty of the methods of at least 3 dB. Comparing $O^{\gamma=0}$ and $O^{W=.25}$ reveals that the differences are smallest in summer when the number of observations is largest (corresponding to the number of reflectivity observations in the two bins adjacent to $Z_e^{\gamma=0}$ and $Z_e^{W=.25}$, Fig. 7.c). Because the method requires a sufficient number of drizzling liquid clouds, we expect that the accuracy of the calibration estimate is reduced in winter. The sensitivity study (Sect. 3.1.1) revealed that higher $N_{tot}$ concen-

5  trations for fixed LWC could lead to higher $Z_e^{\gamma=0}$ and $Z_e^{W=.25}$ values. Assuming that Arctic haze—pollution transported to the Arctic from mid-latitudes—peaks in spring (Shaw, 1995) and leads to enhanced $N_{tot}$ values, the increased $O^{\gamma=0}$ and $O^{W=.25}$ could be related to a change in $N_{tot}$ values. Besides this potentially seasonal impact we cannot identify any trends for NSA. The yearly mean values of 1.8 and 0.1 dB for $O^{\gamma=0}$ and $O^{W=.25}$, respectively, indicate a slight positive calibration offset for NSA. Given the uncertainties, this is in agreement with Kollias et al. (2019) who estimated KAZR's $O$ to be around 3 dB at

10  NSA by using CloudSat observations.

We did not estimate a reference LWP-max($Z_e$) relationship from a model, but given that the KAZR's calibration at NSA is—according to the $\gamma$ and $W$ methods—stable and accurate within 2 dB, we can use the LWP-max($Z_e$) relationship at NSA as a reference. We obtain the reference by taking the mean of the monthly LWP-max($Z_e$) relationships weighted by the number of observations. Based on the average of the yearly mean $O^{\gamma=0}$ and $O^{W=.25}$ values (1.8 and 0.1 dB, respectively), we apply an $O$





value of 1 dB (see Table 4). This allows us to estimate monthly $O^{LWP}$ values from the mean difference between the reference and the corresponding monthly relationship (Fig. 7.a, Table 3). We decided to weight the mean difference by the number of observations in each LWP bin, because the seasonality of the LWP distribution is high and there are only few observations for higher LWP values in winter. Bins with less than 100 observations are skipped. Obviously, this is of limited use for determining

an absolute calibration at NSA, but it allows us to compare the variability of $O^{LWP}$ with $O^{\gamma=0}$ and $O^{W=.25}$. $O^{LWP}$ is varying between $-1.6$ and $1.9$ dB with a standard deviation of $1.1$ dB, which is a reduction by roughly 50% in comparison to $O^{\gamma=0}$ ($2.5$ dB) and $O^{W=.25}$ ($2.0$ dB). Because it is highly unlikely that a variation in the real $O$ would compensate the variability of $O^{LWP}$, but not the variability of $O^{\gamma=0}$ and $O^{W=.25}$, we conclude that the LWP-max($Z_e$) method is the most stable method. The uncertainty of the LWP method is probably half of the two drizzle onset methods (i.e., $1.5$ dB).

Another way to compare the accuracy is to compare the $Z_e$-based relationships after correcting using the various calibration methods. Of course, the variability of $Z_e^{\gamma=0}$ is zero when applying $O^{\gamma=0}$ (Fig. 5.b) and the same applies to $W$ (Fig. 5.g) and LWP (Fig. 5.l). Also, $O^{W=.25}$ leads to a reduction of the variability of $Z_e^{\gamma=0}$ relationship and vice versa (Fig. 5.c, f). This shows the consistency of both methods, but is also related to the fact that the $Z_e^{\gamma=0}$ and $Z_e^{W=.25}$ reference values are close so that both methods use similar sub data sets. When applying, e.g., $O^{LWP}$ to the $Z_e$-$\gamma$ relationship (Fig. 5.d), the variability of

$Z_e^{\gamma=0}$ is not reduced; the inverse operation (applying $Z_e^{\gamma=0}$ to to LWP-max($Z_e$) relation) even enhances the variability (Fig. 5.j). This indicates that the variability of $O^{\gamma=0}$, $O^{W=.25}$ and $O^{LWP}$ is dominated by their intrinsic variability and not by real changes in $O$. This is another indication that $O$ at NSA was very stable in 2016.

## 4.2  Calibration of Oliktok Point (OLI) data

For OLI, the relationships align less well than for NSA: even though most $Z_e$ - $\gamma$ relationships show a quasi-sinusoidal shape,
$Z_e^{\gamma=0}$ varies between approximately -28 dBz and -12 dBz (Fig. 6.a). This is confirmed by the spread of $Z_e^{W=.25}$ (Fig. 6.e) and the LWP-max($Z_e$) relationships (Fig. 6.i) which vary consistently to $Z_e^{\gamma=0}$. The corresponding $O^{\gamma=0}$, $O^{W=.25}$ and $O^{LWP}$ (estimated using NSA as a reference, Table 4) values vary between $-6.9$ and $11.0$ dB (dotted lines Fig. 7.b). There is no reason why the intrinsic variability of the relationships at OLI should be that much higher than at NSA. We conclude that the KAZR at OLI was not properly calibrated, with $O$ likely strongly changing with time. We note that some monthly relationships look

different even after applying $O^{\gamma=0}$, $O^{W=.25}$ and $O^{LWP}$ (estimated using NSA as a reference, Table 4): even after applying a calibration correction (Fig. 6.b, g, l), the spread of the relationships is larger than for NSA. Some months have a drastically reduced amplitude of the $Z_e$-$\gamma$ relationship. Further, many months feature $W > 0.25$ m s$^{-1}$ also for $Z_e < -20$ dBz. Lastly, the LWP-max($Z_e$) relationship is for some months much more curved than the reference relationship. This indicates that the phase classification was not working properly and the data set contains also non-liquid clouds. The phase classification by

Shupe (2007) depends on absolute $Z_e$ values, e.g., by assuming that—under certain conditions—clouds are mixed-phase for $Z_e > -17$ dBz. Consequently, a large positive calibration offset $O$ might result in mixed-phase and ice clouds being falsely classified as liquid clouds because their $Z_e$ value is underestimated. Mixed-phase and ice clouds, however, have different and probably more variable $Z_e$ - $\gamma$, $Z_e$ - $W$, and LWP-max($Z_e$) relationships. A simple solution would be to constrain the data set to cases with temperature larger than 0°C but this is not feasible for Arctic sites because few observations would remain.





**Table 3.** Estimated offsets for NSA and OLI using the three calibration techniques for NSA and OLI. Our best estimate is to use a constant offset of 1 dB for NSA, and to use $O^{LWP}$ for OLI. A positive $O$ value means the $Z_e$ value reported by the radar is too low (Eq. 1).

| | NSA | | | OLI | | |
|---|---|---|---|---|---|---|
| Time | $O^{\gamma=0}$ [dB] | $O^{W=.25}$ [dB] | $O^{LWP}$ [dB] | $O^{\gamma=0}$ [dB] | $O^{W=.25}$ [dB] | $O^{LWP}$ [dB] |
| 2016-01-01 - 2016-01-31 | 2.3 | -1.4 | 0.8 | 9.6 | 7.9 | 7.9 |
| 2016-02-01 - 2016-02-29 | 0.2 | -0.9 | 0.4 | 9.5 | 7.4 | 3.7 |
| 2016-03-01 - 2016-03-31 | 0.2 | -0.8 | -0.5 | 11.7 | 11.1 | 6.3 |
| 2016-04-01 - 2016-04-30 | 4.5 | 3.7 | 0.6 | 9.7 | 6.0 | 2.7 |
| 2016-05-01 - 2016-05-31 | 2.2 | -1.0 | 1.5 | 4.3 | 3.4 | 2.7 |
| 2016-06-01 - 2016-06-15 | 1.1 | -1.4 | -0.5 | 7.1 | 0.9 | 1.1 |
| 2016-06-16 - 2016-07-31 | -2.8 | -2.8 | 0.3 | -5.4 | -7.2 | -4.4 |
| 2016-08-01 - 2016-08-31 | 1.0 | 1.4 | 0.9 | -3.5 | -3.1 | -2.1 |
| 2016-09-01 - 2016-09-30 | 0.8 | 1.4 | 3.0 | -4.3 | -4.6 | -2.3 |
| 2016-10-01 - 2016-10-31 | 2.8 | 0.5 | 1.4 | -0.1 | -1.6 | 1.1 |
| 2016-11-01 - 2016-11-30 | 2.5 | -1.0 | -1.2 | 1.3 | 1.5 | -2.0 |
| 2016-12-01 - 2016-12-31 | 7.2 | 3.4 | 1.6 | 4.3 | -3.4 | 1.9 |
| estimated uncertainty | ±3 | ±3 | ±1.5 | ±3 | ±3 | ±1.5 |

**Table 4.** Reference LWP-max($Z_e$) relationship obtained at NSA using the mean of monthly LWP-max($Z_e$) relationships weighted by the number of observations. Note that the mean $O$ was likely around 1±1 dB for NSA and the reported values in this table are corrected accordingly.

| LWP interval [kg m$^{-2}$ ] | $< \max(Z_e) >$ [dBz] |
|---|---|
| [20, 30[ | -23.35 |
| [30, 40[ | -22.19 |
| [40, 50[ | -21.13 |
| [50, 60[ | -20.60 |
| [60, 70[ | -19.76 |
| [70, 80[ | -19.49 |
| [80, 90[ | -19.35 |
| [90, 100[ | -19.00 |
| [100, 110[ | -18.66 |
| [110, 120[ | -18.40 |

Instead, we run the classification by Shupe (2007) assuming different calibration offsets $O^{\text{phaseclass}}$ from -6 to +10 dBz (with 2 dB steps) and estimate the relationships for every assumed offset. Note that $O^{\text{phaseclass}}$ impacts only *which* data points are





selected based on the phase classification and we do not modify the $Z_e$ values themselves for obtaining $Z_e^{\gamma=0}$, $Z_e^{W=.25}$ and the reference LWP-max($Z_e$) relationship. To obtain a phase classification consistent with the calibration offset, we choose the run with the smallest difference between $O^{\text{phaseclass}}$ and $O^{\gamma=0}$ (or $O^{W=.25}$, $O^{LWP}$). Typically, the smallest difference is less that the 2 dB step size of $O^{\text{phaseclass}}$. After accounting for $O^{\text{phaseclass}}$, the magnitudes of the $Z_e$-$\gamma$ relationships are more

similar (Fig. 8.b), the $W$ for small $Z_e$ values is reduced (Fig. 8.g), and the LWP-max($Z_e$) relationships are less curved (Fig. 8.l). With respect to the used number of observations (Fig. 7.d), application of $O^{\text{phaseclass}}$ reduces the number of observations by approximately half, and makes them more similar to NSA. This indicates that non-liquid clouds have been successfully removed from the data set by accounting for $O^{\text{phaseclass}}$. Interestingly, the differences between $O$ with and without accounting for $O^{\text{phaseclass}}$ are often smaller than 2 dB (Fig. 7.b). This suggests that the methods are more robust than expected and can

provide meaningful calibration estimates even if the liquid cloud data sets are contaminated by non-liquid clouds.

When analyzing $O$ values for OLI, the decrease from June to July 2016 stands out. Even though the decrease magnitude varies between $-5.6$ and $-12.5$ dB, all methods show this decrease (Fig. 7.b, Table 3), and a similar change was reported by (Kollias et al., 2019). Based on discussions with the DOE ARM program, the decrease coincides with a change in the KAZR radar configuration (including the calibration constant) on June 16th 2016, though the details of the change are unclear. To

find out more about this decrease, we also analyze collocated KaSACR measurements. Even though the KaSACR data set size was not sufficient to apply the new calibration methods, we can compare KaSACR and KAZR $Z_e$ measurements directly for the two weeks before and after the step on 2016-06-16. This comparison shows a decrease of the difference between both radars of 7.5 dB (not shown). Because we have no indication for a simultaneous change in the KaSACR's configuration or calibration, we attribute this change to the KAZR confirming the step identified by the liquid cloud methods. The fact that the

relative difference between KAZR and KaSACR was almost zero after 2016-06-16 indicates that the change of the KAZR's configuration was made on purpose to make the measurements of both radars match. After June, all liquid cloud methods show a gradual increase of $O$ with time. Except for December 2016, where less than 1000 observations are available, the agreement of the various methods is high, which indicates that the gradual trend is likely related to the radar and not to the intrinsic variability of the liquid cloud methods. The gradual trend could indicate hardware problems or a dependence of $O$ on the

ambient temperature. The latter could also explain the gradual decrease of $O$ before June 2016. Even though all methods agree about the sign of the trend in spring, $O$ is higher for the $\gamma$ and $W$-based methods than for the LWP method, which is similar to our results for NSA. Therefore, the higher $O^{\gamma=0}$ and $O^{W=.25}$ values could be related to Arctic haze, which has apparently larger impact on $Z_e^{\gamma=0}$ and $Z_e^{W=.25}$ than the LWP-max($Z_e$) relationship. Based on this and on the reduced variability for NSA, we conclude that $O^{LWP}$ is likely suited best for performing an absolute calibration at Arctic sites.

The median difference between $O^{\gamma=0}$ and $O^{W=.25}$ is very similar for OLI and NSA (1.6 and 1.7 dB, respectively) which could indicate a systematic bias between our box model-based estimations of $Z_e^{\gamma=0}$ and $Z_e^{W=.25}$. The fact that the mean $O^{\gamma=0}$ value of 1.8 dB for NSA for 2016 is closest to the 3 dB estimate of Kollias et al. (2019) might suggest that $Z_e^{\gamma=0}$ is closer to reality than $Z_e^{W=.25}$.





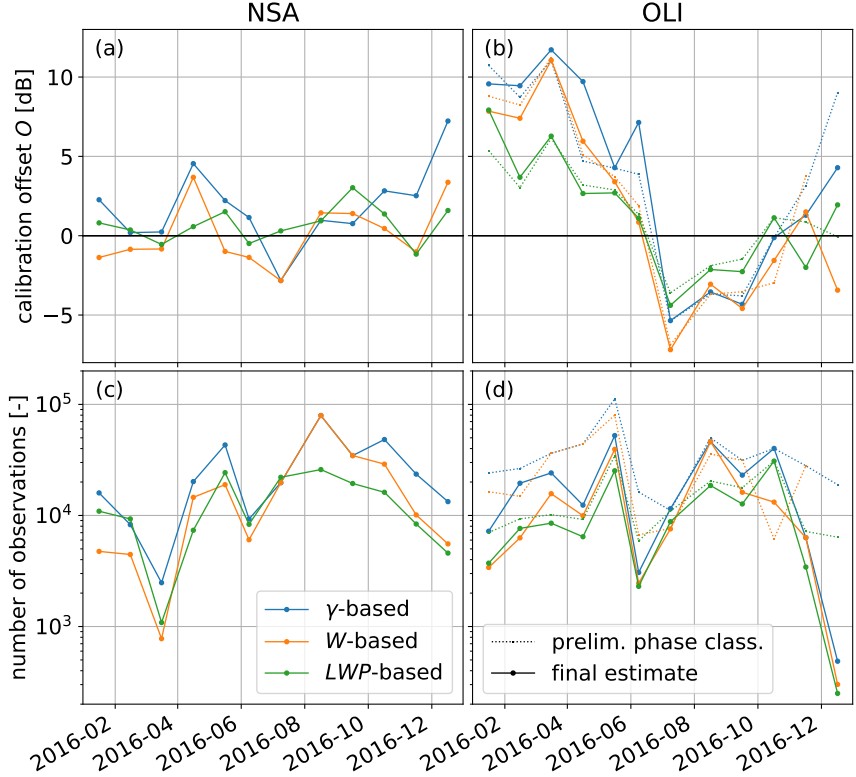

**Figure 7.** Calibration offsets $O$ (a, b) and number of used observations (c, d) for NSA (a, c) and OLI (b, d). For OLI, the dotted lines show the preliminary results without modifying the phase classification with $O^{\text{phaseclass}}$.

### 4.3 Relative calibration of North Slope of Alaska (NSA) and Oliktok Point (OLI) data

The results of the relative high altitude calibration method (Sect. 3.3) are presented in Fig. 9. We found that individual events can bias the statistics when applying the method to monthly periods, therefore we applied the methods to intervals of three months with the bin threshold of July 1st shifted to June 16th. The standard deviation between the different cut-off heights varies between 0.9 and 2.3 dB, which is probably a good estimate for the uncertainty of the method. Assuming that the NSA calibration was stable, the relative comparison reveals the decrease at OLI on 2016-06-16. When using the best cut-off altitudes, the decrease is estimated to be -5.9 dB which is—given the different time intervals used for estimating $O$—in good agreement with the estimate based on the KaSACR-KAZR comparison (-7.5 dB) and the LWP-based method (-5.6 dB).

A comparison of the high altitude cloud method and the new methods is presented in Fig. 10. This requires deriving a relative calibration from $O^{\gamma=0}$, $O^{W=.25}$, and $O^{LWP}$ by subtracting OLI from NSA and averaging the monthly $O$ estimates to





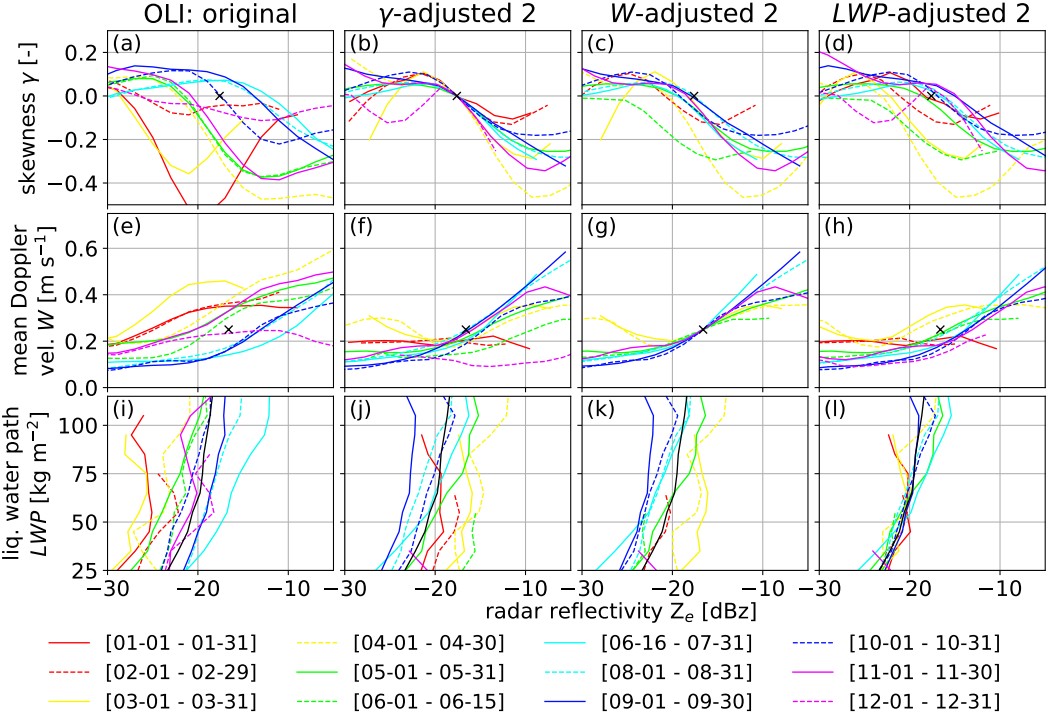

**Figure 8.** As Figure 6, but considering a calibration offset for the phase classification.

three-monthly values. While the combination of both calibrations generally combines the uncertainties of both estimates, some potential error sources cancel out. This is particularly true for any constant or seasonal biases of estimating $Z_e^{\gamma=0}$, $Z_e^{W=.25}$, or the reference LWP-max($Z_e$) relationship. Given the uncertainties, there is excellent agreement (difference $< 3$ dB) between the high altitude and liquid cloud methods for April to December 2016 showing the general feasibility of the liquid cloud

5   methods. For the winter period (Jan - Mar), the agreement is worse, which is likely related to the less robust statistics due to the reduced number of liquid clouds. Moreover, the data sets used for the high altitude method and the liquid cloud methods are not necessarily obtained at the same time even though averaged to the same intervals. This would require the high altitude ice clouds and the liquid clouds to occur always at the same time, which is not the case. In particular when $O$ is shifting quickly, such a temporal mismatch can contribute to the observed differences between the methods. Even though the difference between

10   the LWP-based method and the high altitude clouds method can be up to 2.7 dB in late 2016, the mean difference is lower (0.9 dB) than for the $\gamma$-based (2.0 dB) and $W$-based (1.6 dB) methods. This confirms our previous conclusion that the LWP-based method has the smallest intrinsic variability and likely works best for estimating $O$ in the Arctic.



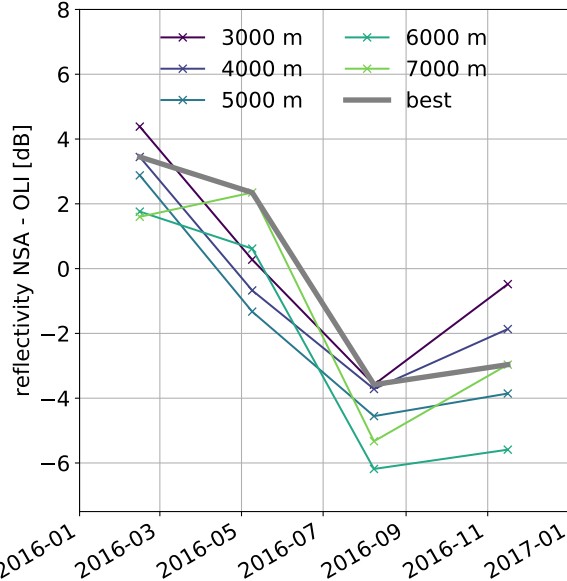

**Figure 9.** Comparison relative calibration offsets NSA - OLI for different cut-off heights (colored lines). The best estimate (see text) is highlighted in gray.

## 5 Summary and conclusions

In this study, we investigate the potential for using the imprint of liquid cloud processes on DSDs for radar calibration. Specifically, we investigate the relationships of *radar reflectively* $Z_e$ to the *skewness* of the radar Doppler spectrum $\gamma$ and to the *mean Doppler-velocity W*. Moreover, we use the relationship between the maximum $Z_e$ value in the column $\max(Z_e)$ and the

5 *liquid water path* LWP measured by a microwave radiometer (MWR). These methods close an important gap in our ability to monitor and assess radar calibration.

The fact that we focus only on drizzle-onset for the $Z_e - \gamma$ and $Z_e - W$ relationships allows us to use a box model (Hoffmann et al., 2017) coupled to the PAMTRA2 radar simulator (Maahn et al., 2015) to determine the dependency of the relationship on the initial DSD and random effects. Depending on initial cloud microphysical conditions and, to a lesser extent, random

effects, we determine typical relationships for $\gamma$ and $W$ as a function of $Z_e$. We find that compensating and saturation effects reduce the variability of the $Z_e - \gamma$ and $Z_e - W$ relations, which allows us to identify reference values of $\gamma$ and $W$ with minimal variability during drizzle-onset. For $\gamma$, we find that $Z_e$ variability is smallest for $Z_e^{\gamma=0} = -17.6 \pm 3$ dBz when cloud droplets and drizzle contribute to reflectivity equally, i.e., $\gamma = 0$. For $W$, we identify the smallest variability for $W = 0.25$ m s$^{-1}$ and $Z_e^{W=.25} = -16.6 \pm 3$ dBz. Because we cannot quantify the impact of feedback effects of clouds and their environment on the

LWP-$\max(Z_e)$ relationship, we do not use a model to obtain a reference relationship. Instead, we use the approach for relative calibration between two radars.





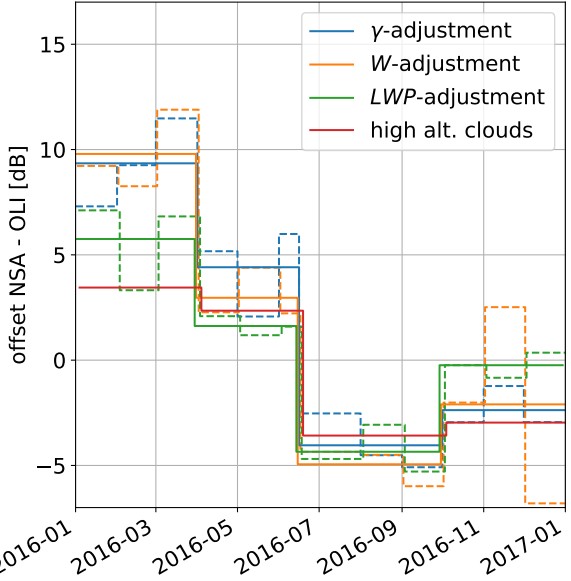

**Figure 10.** Comparison of relative calibrations NSA − OLI using $\gamma$ ($O^{\gamma=0}$, blue line), $W$ ($O^{W=.25}$, orange line), and the high altitude method (green line). All methods have been averaged to the same temporal resolution, the original resolutions are indicated by the dashed lines. For clarity, the lines have been slightly shifted along the x axis.

Applying the methods to radar observations of low-level Arctic liquid clouds at the ARM North Slope of Alaska (NSA) and Oliktok Point (OLI) sites, we identify monthly median $Z_e - \gamma$, $Z_e - W$, and LWP-max($Z_e$) relationships. For NSA, the observed relationships are in general agreement with the box model simulations and we successfully identify the reference $Z_e$ values for $\gamma = 0$ and $W = 0.25$. We use the difference between measured and modeled $Z_e$ reference values for assessing the calibration offset $O$ on a monthly basis. Considering the 3 dB uncertainties of $Z_e^{\gamma=0}$ and $Z_e^{W=.25}$, the calibration at NSA is relatively stable and $O$ is on average around 1 dB (Fig. 7.a). The good calibration of the NSA KAZR motivated us to use the LWP-max($Z_e$) relationship at NSA as a reference for absolute calibration. The variability of the estimated $O^{LWP}$ is smaller than for $O^{\gamma=0}$ and $O^{W=.25}$ indicating that the LWP-based method has an uncertainty of about 1.5 dB and is less impacted by microphysical and dynamical conditions. The difference between the methods is largest for the winter months (Fig. 7.a) indicating that the lower number of liquid clouds might limit the quality of the $O$ estimation. Also, the phase classification algorithm employed might struggle in winter to remove all mixed-phase clouds from the data set as required.

For OLI, we identify serious problems with maintaining an accurate radar calibration. Most remarkably, we find that $O$ decreased 5 to 7 dB in June 2016 (Fig. 7.b), which was likely related to a change in radar configuration even though the details cannot be reconstructed. Further, we identify a slowly decreasing and increasing trend of $O$ in spring and fall, respectively, of 2016. Similar to NSA, the agreement between the liquid-cloud-based methods is reduced during winter indicating that a sufficient number of liquid cloud samples is required for the method to work properly. Despite this, the $Z_e - \gamma$, $Z_e - W$, and




LWP-max($Z_e$) relationships for OLI are consistent after application of a calibration correction (Figs. 6.b,g,l). This indicates the ability of the methods to correct also for larger $O$ values as long as the calibration offset is considered during the phase classification (Shupe, 2007) for identifying liquid clouds. The LWP-based method matches the high altitude cloud reference method best. Considering the error margins, our results are in excellent agreement with Kollias et al. (2019). By applying the

CloudSat method by Protat et al. (2011), they found a similar drop for OLI and a 3 dB offset for NSA.

In summary, we find that the $Z_e - \gamma$, $Z_e - W$, and LWP-max($Z_e$) relationships contain valuable information that can be used to determine the cloud radar calibration offset $O$. Due to the effect of turbulence on radar observations, the $\gamma$ and $W$-based methods likely work best for stratiform clouds, which are typically not that turbulent. In comparison to other calibration methods for cloud radars, the methods have several advantages. Most importantly, no dedicated field campaigns are required

and the method can be easily applied to past data sets. In comparison to the method by (Protat et al., 2011), the liquid cloud microphysical processes methods can be applied to shorter time intervals, which better enables detection of sudden changes. Also, our methods do not depend on CloudSat, which is likely close to the end of its lifetime. Further, the method can be—with limited accuracy in winter—applied to year round observations even at high latitudes because liquid clouds occur throughout the year. The $\gamma$- and $W$-based methods require supporting instrumentation (Microwave radiometer, lidar, radiosonde obser-

vations) only for the identification of liquid clouds. If the presence of ice and mixed-phase clouds can be ruled out by other means (e.g., at sub-tropical or tropical sites), application of the method is possible without any additional instrumentation. The LWP-based method requires a collocated MWR that has to be calibrated carefully using an offset correction during clear-sky periods. While we found the LWP-based method to work best, the question of whether $Z_e^{\gamma=0}$ or $Z_e^{W=.25}$ is the second best method for calibration is still open. The box model indicates a larger stability for $Z_e^{\gamma=0}$, but the variability of observed $Z_e^{W=.25}$

is lower at NSA. Assuming the NSA KAZR calibration was stable, this would indicate that $Z_e^{W=.25}$ is slightly better suited. Yet, $Z_e^{W=.25}$ is more easily affected by biases, e.g., due to persistent vertical air motions related to orography. These biases could be an explanation for the small 1.6 to 1.7 dB offset between the $Z_e^{\gamma=0}$ and $Z_e^{W=.25}$-based calibration estimates. Instead, $\gamma$ is less affected by biases, but observations are more noisy, require a high temporal resolution, and most standard radar products do not include $\gamma$. Likely, it is best to apply all three methods and use the agreement between the methods as an indicator for

the quality of the calibration offset estimate. With respect to the calibration offset $O$ for OLI and NSA in 2016, we recommend using the results of the LWP-max($Z_e$) method for OLI (Fig. 7.b, Table 3) and using an offset of +1 dB for NSA.

Further research is needed to reduce the uncertainty of the methods and to assess the dependence of the reference $Z_e$ values on climatological and environmental conditions like the availability of cloud condensation nuclei. The reference $Z_e$ values and relationships need to be carefully reevaluated when applying the method to radar observations from other regions. This

applies also to the LWP-max($Z_e$) relationship where we used the relationship obtained at NSA as a reference. However, it is not clear whether this relationship is applicable to other sites or whether it is valid only at the North Slope of Alaska. Sites with a radar with stable calibration offsets could be used to assess the seasonality of the used reference relationships over multiple years. Further, an extension of the method to mixed-phase and ice clouds would be desirable, but the greater variability of ice particles shapes, fall velocities, and radar backscattering cross-sections makes this even more challenging than for liquid

clouds. Even though the method has been developed for Ka band cloud radars, it should be generally applicable to zenith



pointing radars using other frequencies. For W-band radars, we expect that the Rayleigh approximation is also mostly valid because—according to the box model—drizzle drops are small enough to assume Rayleigh scattering for $Z_e$ values smaller than -13 dBz. However, one has to correct for attenuation by atmospheric gases and liquid water, which are stronger at W-band.

## 6 Data availability

5  All ARM data products used in the current study are available at the ARM archive www.arm.gov/data. Please refer to the references for the data stream names and DOIs.

*Acknowledgements.* We thank the technicians of the ARM program for operating the instruments in Alaska and Davide Ori for supporting us with the PAMTRA2 development. We acknowledge valuable discussions with Tim Lyons (CU Boulder) and Joseph C. Hardin (PNNL) and thank Graham Feingold (NOAA) for reviewing the manuscript prior submission. This work was supported by the US Department of Energy

10  (DOE) Atmospheric Systems Research (ASR) program (DE-SC0013306, DE-SC00112704) and the NOAA Physical Sciences Division (PSD). This research was performed while F.H. held a Visiting Fellowship of the Cooperative Institute for Research in Environmental Sciences (CIRES) at the University of Colorado Boulder and the NOAA Earth System Research Laboratory.



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
