# Peer review of "Can liquid cloud microphysical processes be used for vertically-pointing cloud radar calibration?"

_Atmospheric Measurement Techniques, 2019_

## Referee Comment (RC1) · Anonymous Referee #1 · 19 Feb 2019

The manuscript is well written but it is tremendously too long and the explanation is fragmented with a very inappropriate amount of details. In all the manuscript there is not a single equation that explain how to perform the radar calibration. To be honest I have to say that I am not familiar with calibration procedures using Cloudsat and ground cloud radar cited by the Authors but I am confident enough with radar calibration in general. However, I feel that the manuscript requires some strong major revision in particular in the presentation form trying to makes the material shorten and giving a procedure that others can follows to calibrate cloud radars. I suggest use appendixes to put in it all the material that is needed to deepen the discussion leaving in the main text the material that go directly to the point.

Major

- Figure 7, which is discussed in the summary and conclusion tells me that the Offset is tremendously variable in time. Even though without considering 5 to 7 dB variation in June 2016 due to radar configuration changes I can see peaks up to 5dBZ from a month to another. Based on this figure the technique proposed seems to be not reliable or hardly applicable also due to the requirements in terms of needed data points.

- While for the skewness vs. Ze and W vs. Ze relationships, you used a model set up able to identify the transition between cloud droplets and drizzle, the Authors have used a different approach when dealing with LWP vs. max(Ze). In this latter case are you sure to consider data samples in the similar environmental conditions as for the previous relationships (i.e. when during cloud to drizzle transition)?

- How long should be the calibration period to have a reliable statistic of the offset?

Minor

pag. 3 line 25, Aquistapace et al., is not accessible at the moment. How yo get her figure 1. pag 5. line 10. Which radiative transfer model (RTM) do you use to convert radiosoundings into brightness temperature and in particular which is the water vapor scheme used by the RTM? Pag 5, line 33, please specify the meaning of singleSIP inizialization method. Pag. 5 and 6, Box model section. Which is the typical time interval needed by the box model to reach its asymptotic state? Is this time consistent with the sampling time of the cloud radar? Pag, 7, figure 1. The explanation of figure 1 can be improved I guess. Everything seems to be compressed and a lot of details a given losing the general sense. In addition, it is not clear to me why you have negative skewness when the spectrum is dominated by drizzle drops (i.e. larger drops than cloud droplets). Pag 15, line 20. The motivation given to do not use the box model to derive LWP-max(Ze) is very unclear to me. Why the LWP vs max(Ze) on one site should be more reliable thatn those at the other site?

---

## Referee Comment (RC2) · Anonymous Referee #2 · 8 Mar 2019

This paper presents three detailed methodologies for the calibration of vertically-pointing millimeter wavelength radars at high latitude ARM sites. The authors make an important and novel contribution which is timely and emerges from an active area of research, has the potential for post-hoc application to many datasets from ARM and other radar deployments—and which may help to reveal significant and sudden changes in radar offsets within those records. The results are well-supported by the methodology, including careful consideration of different sources of uncertainty. The manuscript is well-written and includes a thorough literature review; while the paper runs quite long and requires careful reading, this reviewer does not readily identify any sections which could easily be moved to appendices for the sake of brevity without sacrificing clarity,

but any efforts the authors can make in this regard would be well-spent.

With some attention to the minor comments below, I recommend this paper is accepted for publication.

**1 General comments**

The paper could be made more reader-friendly in two ways:

1. The figures are frequently dense with information and difficult to parse:

   - As a colour-blind reader, reading this paper would be much improved if all figures were drawn with thicker lines and increased panel sizes. This simple change drastically helps distinguish colours, especially as some figures include many lines.

   - Some additional thought is needed to make the key information salient in the figures. In Figs. 5, 6 & 8 the black crosses and black curves would be more easily discerned if they were heavier than the surrounding lines. The black and gray dots demarking the progress of the model in Figs. 1 and 2 may be more easily visible as alternative symbols or vertical lines, or perhaps complemented by labels. If the purple points in Figs. 1 and 2 do not add any information not amply represented by the smoothed orange lines, I suggest they could at least be made lighter or transparent to reduce the clutter in the most important parts of these figures.

2. The narrative of the paper can be difficult to follow. I commend the authors for writing that is precise and free of errors, but especially because this is a detailed (and detail-oriented) paper, the reader would be grateful for additional 'signposts'. As an obvious example, the title of the paper poses a question that could

be much more explicitly answered in both the abstract and the conclusion. Elsewhere, at key points such as the beginnings of the sub-sections of the results, it would help to very clearly state (or repeat) how the previous section motivates what follows. This must be difficult feedback to implement as the manuscript is already relatively long, but just a few well-placed sentences would greatly enhance the readers' experience of this paper. Throughout the methodology and results sections it may also be possible to enhance the narrative by paring away some detail, but I appreciate this is a paper that requires detail.

**2 Specific comments**

- P1, L6: "We identify reference points of these relationship..." Reference points are used only in the skewness and Doppler velocity relationships, not in the LWP method.

- P4, L3: Introducing the calibration offset O in (1), at the end of the paragraph outlining the structure of the paper, seems out of place. O could be just as easily introduced in a brief introduction to Section 3, as it is not referred to in Section 2.

- P26, L22: in considering the differences and offsets between the two drizzle autoconversion calibration methods, would it be helpful to use the box model to estimate the Doppler velocity at drizzle-onset (i.e. $W^{\gamma=0}$), the variability of which might provide some measure of the inherent variability between the two methods?

**3 Typos**

- P6, L11: should be "...controlled..."
- P19, L5: should be "$O^{LWP}$ varies..."

- P19, L7: should be "...compensate for the variability..."

- P21, L4: should be "...less than the 2 dB..."

- P21, L13: should be "...Kollias et al. (2019)."

- P26, L10: should be "...by Protat et al. (2011)."

- P27, L9: should be "...prior to submission."

---

## Author Comment (AC1) · 3 May 2019

**Can liquid cloud microphysical processes be used for vertically-pointing cloud radar calibration? Response to the reviewers**

Maximilian Maahn, Fabian Hoffmann, Matthew D. Shupe,
Gijs de Boer, Sergey Y. Matrosov, Edward P. Luke

April 30th, 2019

*Original Referee comments are in italic*

> manuscript text is indented, with added text underlined and removed text crossed out.

We would like to thank the reviewers for their helpful comments. We revised the manuscript and responded to all of the reviewers' comments.

We evaluated different options to shorten the manuscript, but we did not find a satisfying solution. We could split the manuscript into two papers separating the theoretical study from the applications. We do not find this solution very convincing either, because both manuscripts combined would be even longer due to the need for a second introduction and summary. Therefore, we leave this decision to the editor.

**1  Reviewer I**

*The manuscript is well written but it is tremendously too long and the explanation is fragmented with a very inappropriate amount of details. In all the manuscript there is not a single equation that explain how to perform the radar calibration.To be honest I*

*have to say that I am not familiar with calibration procedures using Cloudsat and ground cloud radar cited by the Authors but I am confident enough with radar calibration in general. However, I feel that the manuscript requires some strong major revision in particular in the presentation form trying to makes the material shorten and giving a procedure that others can follows to calibrate cloud radars. I suggest use appendixes to put in it all the material that is needed to deepen the discussion leaving in the main text the material that go directly to the point.*

We thank the reviewer for his comments. We agree that the manuscript is long, but weren't able to shorten it significantly. To help the reader applying the method to other data sets, we added an appendix with step by step instructions.

**1.1 Major**

*Figure 7, which is discussed in the summary and conclusion tells me that the Offset is tremendously variable in time. Even though without considering 5 to 7 dB variation in June 2016 due to radar configuration changes I can see peaks up to 5dBZ from a month to another. Based on this figure the technique proposed seems to be not reliable or hardly applicable also due to the requirements in terms of needed data points.*

Please note that we estimate the uncertainty of the methods to be 3 dB ($\gamma$, $W$) and 1.5 dB (LWP) For NSA, the standard deviations for the monthly values of the $\gamma$, $W$ and LWP methods are below this, 2.5, 2.0, and 1.1 dB, respectively. It is rather unlikely that potential fluctuations of the calibration and the uncertainty of the method cancel out, therefore we conclude that even the 5 dBz jumps (which can be found only for the $\gamma$ and $W$ methods at NSA) are included by the uncertainty estimate of 3 dB.

It is also important to note that there is no other calibration method available for OLI which allows to capture the changes in short time. Even though the method is far from perfect, the data set would be useless without any correction.

*While for the skewness vs. Ze and W vs. Ze relationships, you used a model set up able to identify the transition between cloud droplets and drizzle, the Authors have used a different approach when dealing with LWP vs. max(Ze). In this latter case are you sure to consider data samples in the similar environmental conditions as for the previous relationships (i.e. when during cloud to drizzle transition)?*

No, we are not sure that the environmental conditions are the same. Actually, the LWP method uses a much wider range of clouds including non-drizzling and drizzling clouds. However, radar calibration does not depend on environmental cloud conditions, so we are confident that this is not a limitation.

*How long should be the calibration period to have a reliable statistic of the offset?*

This is an excellent question. A short calibration period would be impacted by individual events, also the number of liquid cloud observations could be too low. With 15 s temporal resolution, Figure 7 suggests that at least 1000 observations are required. We added to the results:

> We chose monthly intervals as a compromise between the ability to resolve rapid calibration changes and the need for a sufficient number of liquid clouds observations with varying microphysical properties.

and to the summary:

> We applied the methods to monthly intervals to identify rapid changes but obtain a sufficient number of liquid cloud observations (for 15 s temporal resolution, at least 1000 data points).

We agree that a sensitivity study for the ideal period would be desirable, but this would require knowing the true calibration offset at a high temporal resolution.

**1.2 Minor**

*pag. 3 line 25, Aquistapace et al., is not accessible at the moment. How yo get her figure 1.*

We should have provided a copy, but the paper is meanwhile available online.

*pag 5. line 10. Which radiative transfer model (RTM) do you use to convert radiosoundings into brightness temperature and in particular which is the water vapor scheme used by the RTM?*

We added that MWRRET uses MonoRTM (Clough et al., 2005) which we also used to forward model the radiosondes. MonoRTM uses the MT_CKD (Mlawer-Tobin-Clough-Kneizys-Davis) scheme as detailed in the reference.

*Pag 5, line 33, please specify the meaning of singleSIP inizialization method.*

The initialization method has been clarified as follows.

> We apply the so-called *all-or-nothing* approach to calculate collections among the superdroplets  , which has been shown to accurately represent collision-coalescence in the superdroplet
>
>  framework (Unterstrasser et al., 2017). The model is initialized using

the so-called *singleSIP* method (Unterstrasser et al., 2017). In this method, the underlying droplet size distribution is divided into logarithmically spaced bins. Each bin is represented by one superdroplet, which diameter and weighting factor (the number of real droplets represented by  that superdroplet) is determined by integrating the droplet size distribution across the bin. Here, we use 500 bins, i.e., 500 superdroplets to represent the droplet size distribution.

*Pag. 5 and 6, Box model section. Which is the typical time interval needed by the box model to reach its asymptotic state? Is this time consistent with the sampling time of the cloud radar?*

In the box model, collision-coalescence is a runaway process, i.e., without the addition of new droplets, collision-coalescence would continue until only one (very large) droplet is left. Speaking about an asymptotic state is therefore questionable. However, we believe that the reviewer's basic question is valid, asking if we are able to capture the onset of drizzle and its signature in $\gamma$ and $W$ by radar.

For the base case we presented in Sec. 3.1.1, it takes 45 min to reach the maximum in $\gamma$ and another 45 min for the cloud-drizzle balance point at $\gamma \approx 0$ as already stated in the original manuscript. Of course, these timescales depend heavily on the properties of the droplet size distribution, but they indicate the correct order of magnitude which is known to be in the order of tens of minutes for the onset of drizzle. On the other hand, the cloud radar data set used in this study has a temporal resolution of 15 s, which why we are convinced that the considered microphysical processes can be captured successfully. Other studies showed already the potential of observing drizzle onset with cloud radars (e.g., Kollias et al., 2011a,b; Luke and Kollias, 2013; Acquistapace et al., 2019), so that it feels unnecessary to add further verification of our approach.

*Pag, 7, figure 1. The explanation of figure 1 can be improved I guess. Everything seems to be compressed and a lot of details a given losing the general sense. In addition, it is not clear to me why you have negative skewness when the spectrum is dominated by drizzle drops (i.e. larger drops than cloud droplets).*

We extended the discussion of figure 1 with a couple of introductory remarks to guide the reader. Why Skewness becomes negative is now in Section 3.1.1, page 7, line 17ff. Further information about drizzle and skewness can be found in (Kollias et al., 2011a; ?; ?).

*Pag 15, line 20. The motivation given to do not use the box model to derive LWP-max(Ze) is very unclear to me. Why the LWP vs max(Ze) on one site should be more reliable that those at the other site?*

As the word box-model suggests, a box model does not have any spatial dimensions. Therefore, LWP cannot be determined from a boxmodel and we would have to use an

LES model to investigate the impact of $N_{tot}$ on LWP. However, the question how CCN, $N_{tot}$ , and LWP relate on cloud scale is heavily debated (see papers about the Albrecht effect, e.g. **?**) which is why this question is beyond the scope of the study.

Therefore we can use the LWP method only for relative calibration. Only the fact that the methods based on $\gamma$ and $W$ show that the offset at NSA is small and doe snot change with time allow us to use the LWP method also for absolute calibration.

We clarified:

> Focusing only on drizzle-onset has allowed us to use a simple box model to determine the reference points for the $Z_e$-$\gamma$ and $W$ relationships, but addressing the question of how $N_{tot}$ (and the related cloud condensation nuclei concentration) changes LWP cannot be answered with a box model and is beyond the scope of this study.

**2 Reviewer II**

*This paper presents three detailed methodologies for the calibration of vertically pointing millimeter wavelength radars at high latitude ARM sites. The authors make an important and novel contribution which is timely and emerges from an active area of research, has the potential for post-hoc application to many datasets from ARM and other radar deployments and which may help to reveal significant and sudden changes in radar offsets within those records. The results are well-supported by the methodology, including careful consideration of different sources of uncertainty. The manuscript is well-written and includes a thorough literature review but any efforts the authors can make in this regard would be well-spent. With some attention to the minor comments below, I recommend this paper is accepted for publication.*

We thank the reviewer for the positive review.

**2.1 General comments**

*The paper could be made more reader-friendly in two ways:*

*1. The figures are frequently dense with information and difficult to parse: As a colour-blind reader, reading this paper would be much improved if all figures were drawn with thicker lines and increased panel sizes. This simple change drastically helps distinguish colours, especially as some figures include many lines. Some additional thought is needed to make the key information salient in the figures. In Figs. 5, 6 & 8 the black crosses and black curves would be more easily discerned if they were heavier than the surrounding*

*lines. The black and gray dots demarking the progress of the model in Figs. 1 and 2 may be more easily visible as alternative symbols or vertical lines, or perhaps complemented by labels. If the purple points in Figs. 1 and 2 do not add any information not amply represented by the smoothed orange lines, I suggest they could at least be made lighter or transparent to reduce the clutter in the most important parts of these figures.*

We thank the reviewer for this comment. We revised all figures to make them more accessible for color-blind readers.

*2. The narrative of the paper can be difficult to follow. I commend the authors for writing that is precise and free of errors, but especially because this is a detailed (and detail-oriented) paper, the reader would be grateful for additional signposts. As an obvious example, the title of the paper poses a question that could be much more explicitly answered in both the abstract and the conclusion. Elsewhere, at key points such as the beginnings of the sub-sections of the results, it would help to very clearly state (or repeat) how the previous section motivates what follows. This must be difficult feedback to implement as the manuscript is already relatively long, but just a few well-placed sentences would greatly enhance the readers' experience of this paper. Throughout the methodology and results sections it may also be possible to enhance the narrative by paring away some detail, but I appreciate this is a paper that requires detail.*

We agree that more guidance would be helpful and added a couple more signposts. We also removed some redundant information even though this made the manuscript only marginally shorter. We added an appendix with step-by-step instructions on how to apply the method and included a supplement with code examples..

**2.2 Specific comments**

*P1, L6: 'We identify reference points of these relationship...' Reference points are used only in the skewness and Doppler velocity relationships, not in the LWP method.*

We reworded the sentence.

> For each relation, we evaluate the potential for radar calibration.

*P4, L3: Introducing the calibration offset O in (1), at the end of the paragraph outlining the structure of the paper, seems out of place. O could be just as easily introduced in a brief introduction to Section 3, as it is not referred to in Section 2.*

This is an excellent idea, we moved the definition as suggested.

*P26, L22: in considering the differences and offsets between the two drizzle autoconversion calibration methods, would it be helpful to use the box model to estimate the Doppler*

*velocity at drizzle-onset (i.e. $W^{\gamma=0}$), the variability of which might provide some measure of the inherent variability between the two methods?*

This is an interesting idea. We checked our simulations and $W^{\gamma=0}$ is 0.235 m/s which is consistent with the lower $Z_e$ reference value for the $\gamma$ method. However, we think that the inherent variability between both methods can be more easily determined form the $Z_e$ reference values. This is because $Z_e$ is used for binning the observational data and the 1 dB bin spacing shows already that the data sets may partly overlap. Therefore we decided not to inlcude a discussion about $W^{\gamma=0}$.

**2.3 Typos**

*P6, L11: should be '…controlled…'*
*P19, L5: should be 'OLWP varies…'*
*P19, L7: should be '…compensate for the variability…'*
*P21, L4: should be '…less than the 2 dB…'*
*P21, L13: should be '…Kollias et al. (2019).'*
*P26, L10: should be '…by Protat et al. (2011)."*
*P27, L9: should be '…prior to submission.'*

All fixed.

---

## Editor Decision (ED1)

Page 8, line 20:  asses → assess

Page 9, line 1:  is in aggreement → agrees

Page 15, line 1:  this implies the assumption → this assumes

Page 16, line 24: To avoid an impact of attenuation from precipitation →   To avoid precipitation attenuation impacts

Page 18, line 2:  this is in agreement with →  this agrees with

Page 19, line 4:  which is a reduction by roughly 50% → which is a ~50% reduction

Page 24, line 4:  column max (Ze) → column (max(Ze))

---

## Author Response (AR2)

Dear Dr. Kulie,

Thank you for your positive review. We changed the manuscript as suggested.

Regards,
Maximilian Maahn

[revised manuscript text omitted]